# Long-range inhibition from prelimbic to cingulate areas of the medial prefrontal cortex enhances network activity and response execution

Nao Utashiro [1,4], Duncan Archibald Allan MacLaren[1,4], Yu-Chao Liu [1,4], Kaneschka Yaqubi[1,2], Birgit Wojak[1,3] & Hannah Monyer [1] ✉

It is well established that the medial prefrontal cortex (mPFC) exerts top-down control of many behaviors, but little is known regarding how cross-talk between distinct areas of the mPFC influences top-down signaling. We performed virus-mediated tracing and functional studies in male mice, homing in on GABAergic projections whose axons are located mainly in layer 1 and that connect two areas of the mPFC, namely the prelimbic area (PrL) with the cingulate area 1 and 2 (Cg1/2). We revealed the identity of the targeted neurons that comprise two distinct types of layer 1 GABAergic interneurons, namely single-bouquet cells (SBCs) and neurogliaform cells (NGFs), and propose that this connectivity links GABAergic projection neurons with cortical canonical circuits. In vitro electrophysiological and in vivo calcium imaging studies support the notion that the GABAergic projection neurons from the PrL to the Cg1/2 exert a crucial role in regulating the activity in the target area by disinhibiting layer 5 output neurons. Finally, we demonstrated that recruitment of these projections affects impulsivity and mechanical responsiveness, behaviors which are known to be modulated by Cg1/2 activity.

The medial prefrontal cortex (mPFC) exerts modulatory top-down control of diverse behaviors, including performance in attention-demanding tasks[1,2] and pain processing[3,4]. Based on cytoarchitectonic and functional studies, the mPFC has been subdivided in distinct areas, and, at least for some of these areas different nomenclatures have been used in rodent studies[5–8]. In our study we homed in on specific GABAergic projections which in rodents connect the prelimbic (PrL) with the cingulate areas 1/2 (Cg1/2)[9]. The latter is part of the anterior cingulate cortex often referred to as area 24 in primates[7,8]. Notably, there is dense interconnectivity between these and other areas of the mPFC[10].

Given the abundant interconnectivity between mPFC areas, it can be inferred that mPFC areas interact with each other and together shape the net effect of top-down signaling. However, the precise nature of this cross-talk between different mPFC areas is largely unknown. Consistent with such a scenario, recruitment of glutamatergic projections from the PrL to the Cg1/2 attenuates pain-like behavior in mice[11]. In this study, the authors showed that PrL excitatory projections innervated preferentially inhibitory γ-aminobutyric acid-releasing interneurons (GABAergic INs) in the Cg1/2 thereby reducing neuronal activity in the target area. This prompts the question whether another projection between these areas supports the opposite effect, and if so, what is the mechanism thereof.

[1]Department of Clinical Neurobiology at the Medical Faculty of the Heidelberg University and of the German Cancer Research Center (DKFZ), Heidelberg, Germany. [2]Present address: Department of Gastroenterology, Hepatology and Infectious Diseases, University Hospital Düsseldorf and Medical Faculty of Heinrich Heine University Düsseldorf, Düsseldorf, Germany. [3]Present address: Department of Internal Medicine III, University Hospital Ulm, Ulm, Germany. [4]These authors contributed equally: Nao Utashiro, Duncan Archibald Allan MacLaren, Yu-Chao Liu. ✉e-mail: h.monyer@dkfz-heidelberg.de

GABAergic projection neurons are excellent candidates to subserve this role as they are ideally positioned to exert control on neuronal activity in distantly located brain areas for two reasons. Firstly, there is ample anatomical evidence for uni- and bidirectional cortico-cortical GABAergic projections within and across hemispheres[12–18]. Secondly, functional studies of cortico-subcortical GABAergic projections revealed a leitmotiv regarding their connectivity[19,20]. Specifically, several studies provided electrophysiological evidence that subcortical-cortical GABAergic projections inhibited local GABAergic INs in the target area, thereby causing disinhibition of the circuits that these GABAergic INs are embedded in[21,22]. It is tempting to speculate that cortico-cortical GABAergic projections are subject to the same rule of connectivity. This is indeed the case as we demonstrate in this study homing in on two areas of the mPFC that are connected via GABAergic projection neurons.

Thus, based on anterograde and retrograder tracing experiments, we identified GABAergic projection neurons connecting the PrL and the Cg1/2. Electrophysiological recordings in acute slices allowed us to establish the identity of the targeted cells and calcium imaging investigations in vivo enabled us to study the effect of GABAergic projection neuron recruitment on ensemble activity in the Cg1/2. Together these investigations aided us in determining a mechanism by which PrL-derived GABAergic projections modulate neuronal activity in the Cg1/2, as also evidenced in several experimental paradigms at the behavioral level.

## Results

### GABAergic projections from the PrL to the Cg1/2 target L1 GABAergic INs

To identify potential GABAergic projections between the PrL and the Cg1/2, we injected a Cre-dependent adeno-associated virus (AAV-CAG-Flox-eGFP-WPR6) to drive Cre-dependent expression of enhanced green fluorescent protein (eGFP) in the PrL of $GAD67^{+/Cre}$ (henceforth termed $GAD^{Cre}$) mice[23]. This resulted in labeling of PrL GABAergic neurons and revealed projections in the Cg1/2. Strikingly, axons of PrL GABAergic projection neurons were located mainly in layer 1 (L1) of the Cg1/2.

To quantitatively analyze the laminar distribution of GABAergic axons in the target area, and see how this compares with that of glutamatergic axons, we co-injected AAV-CAG-Flox-eGFP-WPR6 and AAV-CaMKIIa-hChR2-mCherry into the PrL of $GAD^{Cre}$ mice (Fig. 1a–f). The density of PrL GABAergic projection neuron-derived axons was indeed significantly higher in L1 compared to that of other layers (L1 vs. each layer, $p < 0.05$, two-sided Dunnett's test, 9 brain slices from $N = 3$ mice; Fig. 1g, h, k, m). In contrast, the density of glutamatergic axons was comparable between layers (Fig. 1g, i, l, n). Within L1, GABAergic axons were denser in L1a compared to L1b ($p < 0.01$, one-sample $t$-test of the percentage of GABAergic axons in L1a versus 50%; Fig. 1j, k, o). This was not the case for glutamatergic axons (Fig. 1j, l, o).

To determine the location of GABAergic projection neuron cell bodies within the PrL, we performed retrograde tracing experiments (Supplementary Fig. 1). Following cholera toxin subunit B conjugated to Alexa Fluor 555 (CTB-Alexa 555) injection into the Cg1/2 of $GAD67^{EGFP}$ mice ($N = 3$ mice; Supplementary Fig. 1a, b), we detected retrogradely labeled $CTB^+/GFP^+$ neurons in all layers of the PrL (Supplementary Fig. 1c–g).

To reveal the molecular identity of GABAergic projection neurons, we performed tracing experiments homing in on the three major GABAergic cell types in the neocortex, that is somatostatin positive ($SOM^+$), parvalbumin positive ($PV^+$), and 5-hydroxytryptamine receptor 3A positive ($5HT_{3A}^+$) neurons[24–27]. The latter comprise vasointestinal peptide positive ($VIP^+$) and $VIP^-$ neurons[28]. To this end, we injected AAV-CAG-Flox-eGFP-WPR6 into the PrL of $SOM^{Cre}$, $PV^{Cre}$ and $VIP^{Cre}$ mice and CTB-Alexa 555 into the Cg1/2 of $5HT_{3A}^{EGFP}$ mice (Supplementary Figs. 2 and 3). We identified axons of PrL GABAergic projection neurons in the Cg1/2 of $SOM^{Cre}$ ($N = 3$ mice; Supplementary Fig. 2a–d), but not of $PV^{Cre}$ ($N = 2$ mice; Supplementary Fig. 2e–h) or $VIP^{Cre}$ mice ($N = 5$ mice; Supplementary Fig. 3a–d). Furthermore, we found $CTB^+/GFP^+$ neurons in the PrL of $5HT_{3A}^{EGFP}$ mice ($N = 3$ mice; Supplementary

Fig. 3e–j). Based on these results we conclude that PrL GABAergic projection neurons comprise $SOM^+$ and $5HT_{3A}^+/VIP^-$ neurons.

As GABAergic projection neurons often connect two brain areas or regions bidirectionally[19,29], we directly tested whether this is also the case for the mPFC areas. Indeed, based on virus-mediated tracing, we identified GABAergic projections also in the opposite direction, that is from the Cg1/2 to the PrL (Supplementary Figs. 4 and 5). Together these results indicate that GABAergic projections connect the PrL and the Cg1/2 reciprocally, and the two projections bear a striking resemblance regarding the axon localization, which is preferentially in L1a (compare Fig. 1m, o with Supplementary Fig. 4m, o). Notably, GABAergic projection neurons connecting different cortical areas do not seem to be confined to the mPFC as we identified GABAergic neurons in the PrL projecting to L1 of the insular cortex (Supplementary Fig. 6).

We next sought to establish the identity of cells that are targeted by PrL GABAergic projection neurons. To this end, AAV-DIO-ChR2-mCherry was injected into the PrL of $GAD^{Cre}$ mice, and whole-cell patch-clamp recordings were performed from neurons located in L1 and layer 2/3 (L2/3) of the Cg1/2. Using a high $Cl^-$ internal solution, optic fiber-guided laser stimulation (5-ms pulses, 473-nm wavelength) of ChR2-expressing axons evoked inward currents in L1 INs of the Cg1/2. Based on their firing pattern, L1 INs could be further classified as putative single-bouquet cells (pSBC) and neurogliaform cells (pNGF)[30] (Fig. 2a, Supplementary Table 1 and Supplementary Fig. 7). Reconstruction of biocytin-filled neurons revealed distinct morphologies, in particular when considering the extension of axonal arborization along the horizontal and vertical axes (Fig. 2a and Supplementary Fig. 8). The optically evoked currents were almost fully abolished in the presence of the $GABA_AR$ blocker SR95531 (SR, 1 µM; $0.5 \pm 0.4\%$ of control, $n = 5$ cells, $N = 3$ mice; Fig. 2a). Notably, optically evoked inhibitory postsynaptic currents (IPSCs) were not detected in other electrophysiologically defined neurons in L2/3, but only in L1 INs of the Cg1/2 (IPSC peak amplitude: pNGF, $28.96 \pm 10.15$ pA, $n = 16$ cells, pSBC, $51.2 \pm 17.79$ pA, $n = 11$ cells; connectivity: pNGF, 36.21%, 21 of 58 cells, pSBC, 52.78%, 19 of 36 cells, $pSOM^+$, 0 of 10 cells, $pPV^+$, 0 of 29 cells, $p5HT_3R^+$, 0 of 11 cells, L2/3 PN, 0 of 34 cells, $N = 22$ mice; Fig. 2b). Interestingly, we found the same rule of connectivity in the opposite direction. Thus, GABAergic projection neurons from the Cg1/2 to the PrL also inhibit selectively pSBC and pNGF in L1 (IPSC peak amplitude: pNGF, $18.62 \pm 10.48$ pA, $n = 4$ cells, pSBC, $10.97 \pm 2.97$ pA, $n = 9$ cells; connectivity: pNGF, 8.33%, 4 of 48 cells, pSBC, 36%, 9 of 25 cells, $pSOM^+$, 0 of 14 cells, $pPV^+$, 0 of 9 cells, $p5HT_3R^+$, 0 of 12 cells, L2/3 PN, 0 of 52 cells, $N = 22$ mice).

To investigate the effect of PrL GABAergic projections on the excitability of L1 INs in the Cg1/2, we performed cell-attached and whole-cell recordings in L1 INs. We evoked suprathreshold responses by placing the stimulating electrode in the L1 region comprising the long-range GABAergic axons en route to the target region (Fig. 2c). The spiking probability markedly decreased both in pSBCs and pNGFs when the optical stimulation preceded electrical stimulation (E-stim) by 5 ms (pSBC: without optical stimulation, $73.33 \pm 3.09\%$, with optical stimulation, $44.76 \pm 6.17\%$, after optical stimulation, $60.48 \pm 5.46\%$, $n = 7$ cells, $N = 4$ mice; pNGF: without optical stimulation, $48.67 \pm 4.55\%$, with optical stimulation, $22.67 \pm 6.78\%$, after optical stimulation, $44.00 \pm 8.19\%$, $n = 5$ cells, $N = 3$ mice; $p < 0.005$, $p < 0.05$, Friedman test followed by post hoc two-sided Dunn's test; Fig. 2d, e). This result suggests an inhibitory/shunting GABAergic action of PrL projections on L1 INs in the Cg1/2, and implies a potential disinhibitory effect in the Cg1/2 network[31]. To directly test the net effect of PrL GABAergic projections on layer 5 pyramidal neurons (L5 PNs), as they are the output cells in cortical circuits, we performed whole-cell recordings in L5 PNs with a low $Cl^-$ internal pipette solution and evoked postsynaptic currents by stimulating in L1 (Fig. 2f). In addition, we tested the potential connectivity of PrL GABAergic projections with L5 PNs. We found no evidence for connectivity between PrL GABAergic

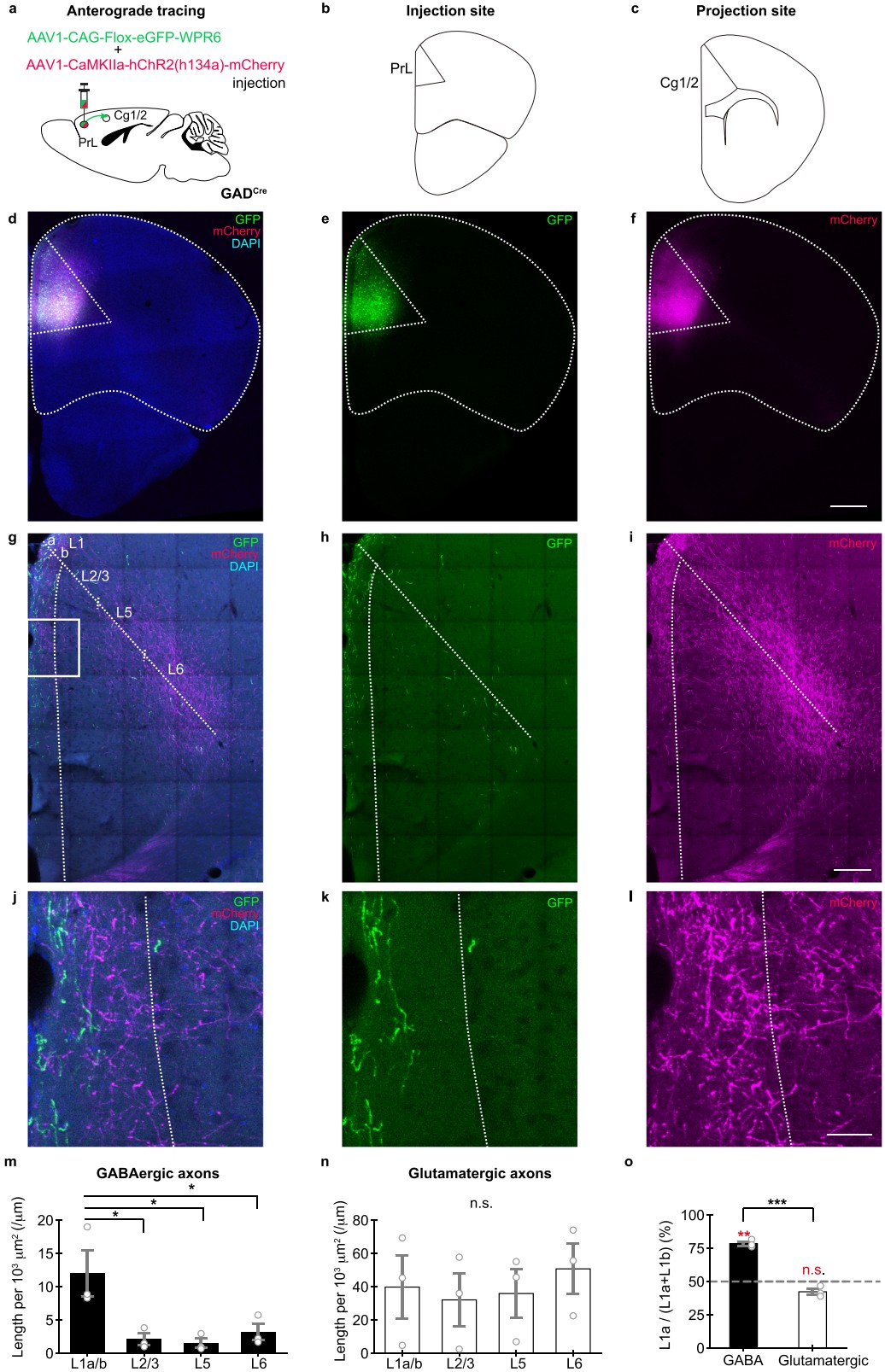

**m** GABAergic axons  **n** Glutamatergic axons  **o**

projections and L5 PNs (0 of 17 cells; Fig. 2b). We recorded electrically evoked compound PSCs by clamping the voltage at near −50 mV and putative pure excitatory postsynaptic currents (EPSCs) at −92 mV, the equilibrium potential of ionotropic GABA receptors ($E_{GABA}$; Fig. 2g, see "Methods"). Compared to electrical stimulation alone, the inward charge recorded at near −50 mV increased by 42.3% when the optical stimulation preceded electrical stimulus by 5 ms (E-stim alone,

0.202 ± 0.046 pC, E-stim with optical stimulation, 0.287 ± 0.061 pC, $n = 12$ cells, $N = 4$ mice, $p < 0.005$, two-sided Wilcoxon signed rank test; Fig. 2g–i). The increased inward charge likely reflects reduced feed-forward IPSCs. To isolate the effect on IPSCs, we recorded the cells at −92 mV ($E_{GABA}$). As expected, the inward charge remained unchanged (E-stim alone, 0.667 ± 0.120 pC, E-stim with optical stimulation, 0.640 ± 0.115 pC, $n = 10$ cells, $N = 4$ mice, $p = 0.106$, two-sided Wilcoxon

**Fig. 1 | PrL GABAergic projection neurons target mainly L1 in the Cg1/2.**
**a** Schematic of the injection for anterograde tracing. AAV1-CAG-Flox-eGFP-WPR6
and AAV1-CAMKIIa-hChR2(h134a)-mCherry were injected into the PrL of GAD^Cre
mice. **b** Schematic of the injection site into the PrL and **c** the projection site in the
Cg1/2 in a coronal section. **d** Representative confocal image of Cre-dependent eGFP
(green) and CAMKII-promoter dependent mCherry (magenta) expression at the
injection site in the PrL in a DAPI stained (blue) coronal section. **e** Cre-dependent
eGFP expression in GABAergic neurons. **f** mCherry expression in excitatory neu-
rons. **g** Representative confocal image of a DAPI stained coronal section showing
eGFP and mCherry at the projection site, i.e., the Cg1/2. **h** PrL GABAergic neuron-
derived GFP expressing axons in the Cg1/2. **i** PrL excitatory neuron-derived
mCherry-expressing axons in the Cg1/2. **j–l** Magnified view of the boxed area in (**g**).
**m** Density of GABAergic axons (L1 vs. L2/3, $p = 0.0166$, L1 vs. L5, $p = 0.0122$, L1 vs. L6,

$p = 0.0297$, DF = 8, two-sided Dunnett's test) and **n** excitatory axons in the indicated
layers of the Cg1/2 (L1 vs. L2/3, $p = 0.974$, L1 vs. L5, $p = 0.996$, L1 vs. L6, $p = 0.932$,
DF = 8, two-sided Dunnett's test). Open circles represent mean values from indivi-
dual mice. **o** GABAergic and excitatory axons calculated as percentage of total
axons in L1a ($p = 0.000216$, DF = 4, two-sided $t$-test). Red asterisks and n.s. above
bars refer to a one-sample $t$-test of the percentage of GABAergic or excitatory
neuron axons in L1a versus 50% (GABAergic, $p = 0.00337$, DF = 2, glutamatergic,
$p = 0.0783$, DF = 2). **d–o** Images and data were obtained from nine slices from three
mice. **m–o** Data are presented as mean ± S.E.M. Scale bar in (**d–f**), 500 μm, in (**g–i**),
200 μm, in (**j–l**), 50 μm. PrL prelimbic cortex, Cg1/2 cingulate area 1 and 2, eGFP
enhanced green fluorescent protein, DAPI 4′,6-diamidino-2-phenylindole. n.s. not
significant, *$p < 0.05$, **$p < 0.01$, ***$p < 0.001$, DF degree of freedom.

signed rank test; Fig. 2g–i). The normalized inward charge was also
significantly larger in the recordings performed at near −50 mV com-
pared to that in the recordings at −92 mV (near −50 mV: 1.446 ± 0.125,
−92 mV: 0.947 ± 0.018, $n = 10$ cells, $N = 4$ mice, $p < 0.005$, Wilcoxon
signed rank test; Fig. 2j). Taken together, these results demonstrated
that the GABAergic projections from the PrL selectively innervate L1
INs in the Cg1/2 and disinhibit L5 PNs possibly by reducing feedforward
inhibition from L1 INs.

### Stimulation of PrL GABAergic projections in the Cg1/2 increases impulsivity in a visual attention task

Given the involvement of the Cg1/2 in attentional processes[32–34] and
the well-studied function of the PrL in top-down control of multiple
cortical and subcortical areas, including the Cg1/2[35–37], we tested
whether GABAergic projections from the PrL to the Cg1/2 influence the
performance of mice in an operant test of sustained attention[38,39]. In
this test, the animals are required to rapidly respond to a brief increase
in the intensity of a stimulus light by pressing a lever (Fig. 3). Mice
could make four types of responses: dark—pressing when there was no
illumination; premature—pressing during the anticipatory stage where
there was dim illumination prior to the brief increase in illumination;
correct—pressing during the brief period of full illumination (which led
to the delivery of a reward pellet); and omission—not pressing the lever
at all (Fig. 3b). Pressing at any point other than during the correct
period led to the end of the current trial without reward delivery. The
attentional demands of the task were manipulated by decreasing the
difference between the illumination level during the premature stage
versus the correct stage. In order to activate PrL GABAergic terminals
in the Cg1/2, AAV-DIO-ChR2-mCherry (ChR2 group, $N = 9$ mice) or, as
control, AAV-DIO-YFP (YFP group, $N = 10$ mice) was injected into the
PrL of GAD^Cre mice, and an optic fiber was implanted into the Cg1/2
(Fig. 3a). During the sham condition, i.e., when mice were connected to
the optic fiber, but no light was delivered, there was no difference
between the two groups. There was also no difference between the YFP
and ChR2 group during optical stimulation (10-ms pulses at 20 Hz, 473-
nm wavelength, ~4-mW intensity delivered during the entire session)
during the low demand version of the task (YFP group vs. ChR2 group:
all $p$ values for correct responses, premature responses omissions, and
dark responses > 0.10) (Fig. 3c–e top panels). In contrast, during the
high demand version of the task, stimulation caused a significant
decrease in correct responses (YFP group vs. ChR2 group: $p = 0.0035$,
RM ANOVA) (Fig. 3c bottom panel) and a corresponding increase in
premature responses ($p = 0.0085$, RM ANOVA) (Fig. 3d bottom panel)
but no significant change in omissions (Fig. 3e bottom panel, $p > 0.10$)
or dark responses ($p > 0.10$), indicating increased impulsivity.

### Optogenetic activation of PrL GABAergic projections in the Cg1/2 increases the response to mechanical stimulation

Next, we tested whether in another sensory modality response
execution is altered upon recruitment of GABAergic PrL projections in
the Cg1/2 by combining optogenetic stimulation with a mechanical

stimulation assay. In this assay, mice receive plantar applications of
graded mechanical stimuli (von Frey filaments) exerting forces
between 0.04–1.0 g and one assesses whether the mouse withdraws
his paw in response to the stimulus (Fig. 4). We injected AAV-DIO-
ChR2-mCherry ($N = 9$ mice) or AAV-DIO-YFP as a control ($N = 9$ mice)
into the PrL of GAD^Cre mice, and optically stimulated the terminals in
the Cg1/2 via an implanted optic fiber (Fig. 4a). Mechanical sensitivity
of the hind paw was assessed in test sessions prior to, during, and
following optical stimulation (10-ms pulses at 20 Hz, 473-nm light
delivered during the entire test session). There was no difference in
basal mechanical sensitivity between ChR2 and YFP groups prior to
optical stimulation (Fig. 4b). However, during optical stimulation of
PrL GABAergic terminals in the Cg1/2 there was an increased sensitivity
to the 0.16-g and 0.4-g filaments (Fig. 4c, $p < 0.0001$ and $p = 0.0003$
respectively, RM ANOVA with two-sided Bonferroni multiple compar-
ison) which returned to normal once the stimulation was terminated
(Fig. 4d). This left-shift of the stimulus-response curve indicates a
higher responsiveness upon recruitment of GABAergic PrL projections
in the Cg1/2.

### Optogenetic activation of PrL GABAergic projections in the Cg1/2 increases capsaicin-evoked mechanical hypersensitivity

Given that the Cg1/2 has been implicated in nociceptive responses in
rodents and pain perception in humans[40–42], we assessed whether L1 INs
in Cg1/2 are involved in a capsaicin-evoked model of mechanical hyper-
sensitivity. This test involves capsaicin injection into the lower hind leg
and application of von Frey filaments to the plantar hind paw. Indeed,
analysis of cFos activity revealed a higher number of cFos⁺ L1 INs in Cg1/2
of mice that received the treatments ($N = 6$ mice) compared to control
mice without capsaicin injection and von Frey filament application ($N = 6$
mice, Supplementary Fig. 9a, b, $p = 0.0348$, RM ANOVA).

We next tested whether activation of PrL GABAergic neuron
projections in the Cg1/2 affects mechanical sensitivity of mice exposed
to this paradigm. Thus, we injected AAV-DIO-ChR2-mCherry ($N = 9$
mice) or AAV-DIO-YFP as a control ($N = 9$ mice) into the PrL of GAD^Cre
mice and optically stimulated the terminals in the Cg1/2 via an
implanted optic fiber (Supplementary Fig. 9c). There was no difference
in basal mechanical sensitivity between ChR2 and YFP groups prior to
capsaicin injection and optical stimulation (Supplementary Fig. 9d).
However, in the test session 15 min after capsaicin injection and during
optical stimulation of PrL GABAergic terminals in the Cg1/2 (10-ms
pulses at 20 Hz, 473-nm light delivered during the entire test session)
there was an overall increased sensitivity to application of von Frey
filaments (Supplementary Fig. 9e, $p = 0.0069$, RM ANOVA). Further-
more, this increased sensitivity persisted in the test session 15–30 min
after the termination of optical stimulation (Supplementary Fig. 9f,
$p = 0.0042$, RM ANOVA). This left-shift of the stimulus-response curve
mirrors the higher responsiveness upon recruitment of GABAergic PrL
projections in the Cg1/2 that was observed in the absence of capsaicin
during optogenetic stimulation (Fig. 4), but differs in that it continues
during the post-stimulation period.

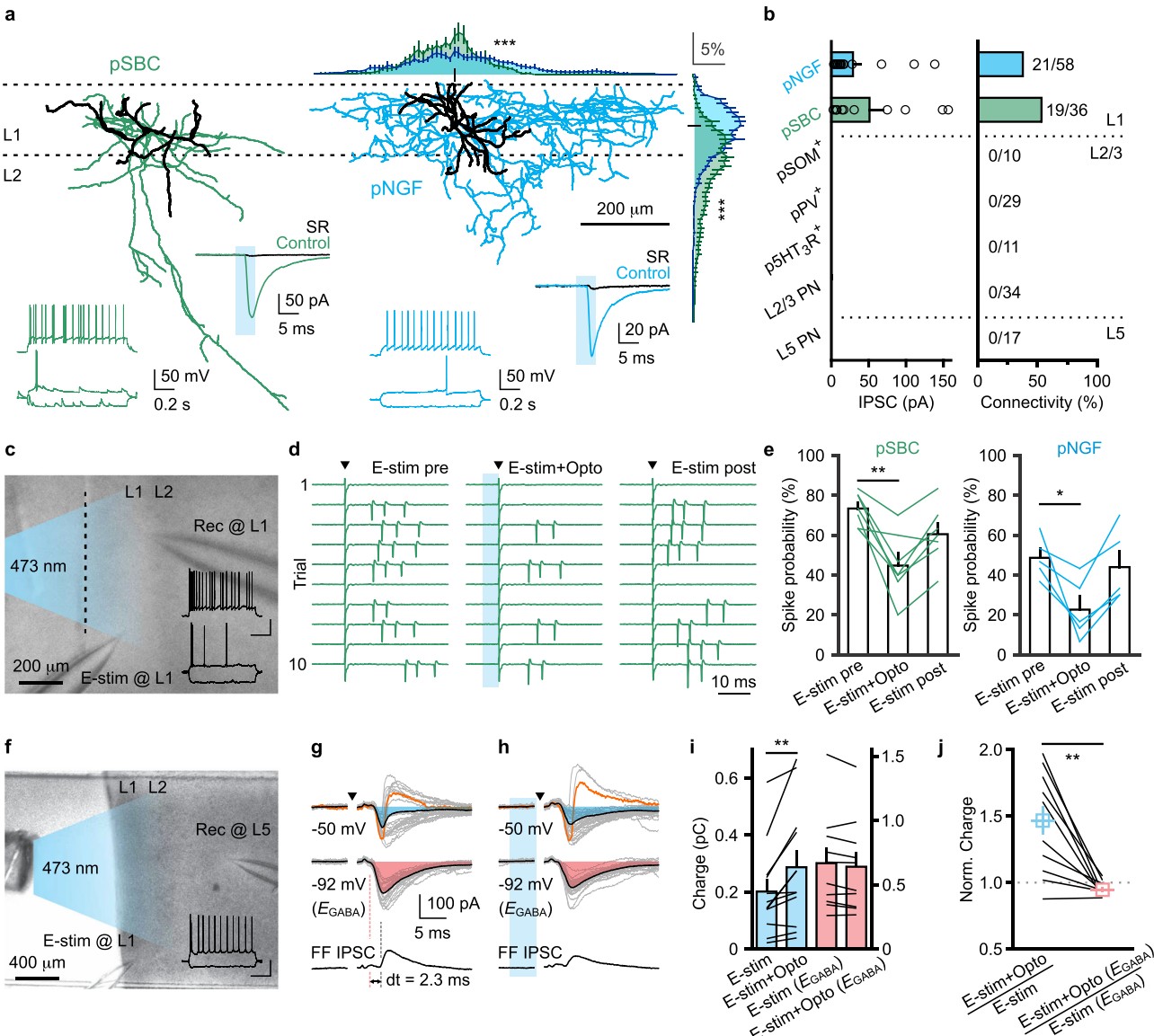

**Fig. 2 | PrL GABAergic projection neurons preferentially target L1 INs and disinhibit L5 PNs in the Cg1/2. a** Reconstructed pSBC (left) and pNGF (right) responsive to axonal stimulation of GABAergic projections. Somata and dendrites are indicated in black. Axonal arborization is indicated in green and blue. Normalized axonal density for pSBC ($n = 6$ cells, $N = 3$ mice) and pNGF ($n = 7$ cells, $N = 3$ mice) plotted against horizontal or vertical axes; measurements were performed in 10 μm intervals (black tick indicates location of the soma, horizontal axis: $F(160,1760) = 2.673$, vertical axis: $F(160,1760) = 7.865$, ***$p < 0.0001$, two-way RM ANOVA). Inset, typical firing patterns in response to current injections (−50, near threshold and 50 pA) and corresponding light-evoked IPSCs. Blue columns indicate laser illumination. Note that the light-evoked IPSCs are almost completely abolished ($n = 5$ cells, $N = 3$ mice) by the GABA$_A$R blocker SR95531 (SR, 1 μM). **b** Summary of IPSC peak amplitude (left; pNGF, $n = 16$ cells, pSGC, $n = 11$ cells) and connectivity (right). The number of connections per number of attempts for each cell type is indicated ($N = 22$ mice). **c** Experimental setting in a coronal section. Stimulating electrode is placed in L1. GABAergic axon stimulation occurs via blue light. The firing pattern belongs to a pSBC in L1, the approaching recording pipette (rec) is visible on the right. Pial surface is indicated as a dotted line. Inset, the firing patterns of the recorded pSBC. Scale bar, 40 mV, 0.25 s. **d** Cell-attached recording of the same pSBC shown in (**c**). Left and right, action currents induced by electrical stimulation (E-stim, triangle) only. Middle, optical stimulation (Opto, blue column) precedes electrical stimulation by 5 ms. **e** Summary of spiking probability of pSBC ($n = 7$ cells, $N = 4$ mice, $p = 0.0025$) and pNGF ($n = 5$ cells, $N = 3$ mice, $p = 0.0342$) before, during and after optical stimulations. **$p < 0.005$, *$p < 0.05$, Friedman test

followed by post hoc two-sided Dunn's test. **f** Experimental setting for whole-cell recording of L5 PNs. Stimulating electrode is placed in L1, GABAergic axons are activated by blue light. Inset, the firing pattern of a recorded PN. Scale bar, 40 mV, 0.25 s. **g**, **h** Electrically evoked responses from the same PN shown in (**f**). Top, compound PSCs (including EPSC-IPSC sequences and putative pure EPSCs) recorded at −50 mV. Individual traces are superimposed and shown in gray. Averaged traces are shown in black. Two EPSC-IPSC sequences are highlighted in orange. Shaded areas indicate the inward charge. Note that the stimulation not always elicited EPSC-IPSC sequences as the spike probability of L1 INs in (**d**, **e**) was always below 100%. Middle, EPSCs recorded at −92 mV ($E_{GABA}$). Bottom, feedforward (FF) IPSCs isolated by subtraction (see "Methods"). The onset of the EPSC and of the FF IPSC are indicated in red and black dotted line, respectively. In (**h**), optical stimulation (blue column) precedes electrical stimulation (triangle) by 5 ms. Stimulation artifacts were removed for clarity sake. **i** Summary of inward charge recorded at near −50 (blue, $n = 12$ cells, $p = 0.0015$) or −92 mV ($E_{GABA}$, red, $n = 10$ cells, $p = 0.1055$) with and without optical stimulation ($N = 4$ mice). **$p < 0.005$, two-sided Wilcoxon signed rank test. **j** Summary of normalized inward charge recorded at near −50 (blue) or −92 mV (red) ($n = 10$ cells, $N = 4$ mice, $p = 0.0039$, **$p < 0.005$, two-sided Wilcoxon signed rank test). **b**, **e**, **i**, **j** Data are presented as means ± S.E.M. pSBC putative single-bouquet cell, pNGF putative neurogliaform cell, pSOM$^+$ putative somatostatin-expressing cell, pPV$^+$ putative parvalbumin-expressing cell, p5HT$_3$R$^+$ putative serotonin 3A receptor-expressing cell, PN pyramidal neuron, IPSC inhibitory postsynaptic current, EPSC excitatory postsynaptic current.

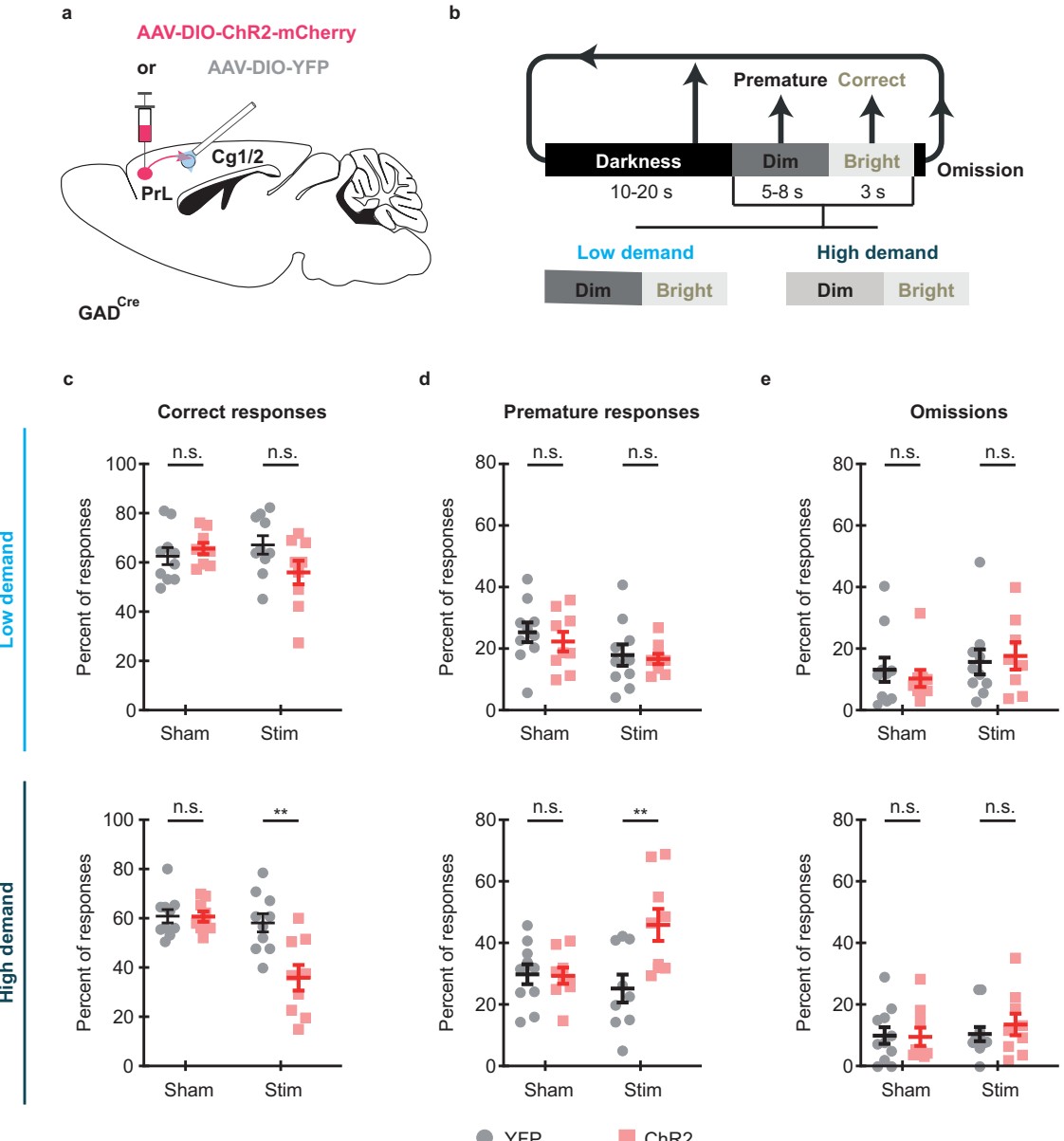

**Fig. 3 | Optogenetic activation of of PrL GABAergic projections in the Cg1/2 increases impulsivity. a** AAV-DIO-ChR2-mCherry (ChR2 group, *N* = 9 mice) or AAV-DIO-YFP (YFP control group, *N* = 10 mice) was injected into the PrL of GAD<sup>Cre</sup> mice and an optic fiber was implanted into the Cg1/2. **b** Schematic of the operant task. Mice were trained in an operant task where a briefly illuminated [3 s] bright light signaled that pressing a lever would lead to delivery of a reward. The illumination of the bright light was preceded by a dim light for an unpredictable duration [5–8 s] during which time lever pressing was punished with darkness [10–20 s]. Therefore, the mice had to attend to the light and, in order to earn a food pellet reward, rapidly respond when the intensity increased. The attentional demands of the task could be manipulated by changing the intensity of the dim light. Thus, in the low demand version of the task there was a greater difference between the dim and bright light states compared to the high demand version of the task. The task has four response types: Correct responses = lever pressing during the bright light state; premature responses = lever pressing during the dim light state; omissions = failure to lever press during either the dim or bright light states and dark = pressing when the stimuli light was not illuminated. Once trained to ~60% correct responses, mice were tested

in a sham session (Sham) where no light was delivered into the Cg1/2 and a stimulation session (Stim) where blue light pulses (10-ms pulses at 20 Hz, 473-nm wavelength) were delivered during the entire session. **c** Correct responses during sham and stimulation conditions. Stimulation has no effect during the low demand task (top, all *p* > 0.05), but significantly reduces correct responses during the high demand task (bottom, *group × demand* interaction *F*(3,49) = 5.237, *p* = 0.0032) with YFP vs. ChR2 in the high demand task with stimulation *p* = 0.0035). **d** Premature responses during sham and stimulation conditions. Stimulation has no effect during the low demand task (top, *p* > 0.05), but significantly increases premature responses during the high demand task (*group × demand* interaction *F*(3,49) = 5.489, *p* = 0.0025, with YFP vs. ChR2 in the high demand task with stimulation *p* = 0.0085) indicating increased impulsivity. **e** Stimulation has no significant effect on omissions during either the low or high demand versions of the task (all *p* > 0.05). All data were analyzed with Geisser-Greenhouse corrected RM ANOVA followed by two-sided multiple comparisons using the two-stage method of Benjamini, Krieger and Yeku-tieli. Lines on graphs show group means ± S.E.M., dots (YFP) and squares (ChR2) show individual data points. **\*\****p* ≤ 0.01.

## Optogenetic activation of PrL GABAergic projections enhances stimulus-evoked responses of excitatory cells in the Cg1/2

To directly test how PrL GABAergic neurons affect the activity in the Cg1/2, we performed two-photon calcium (Ca²⁺) imaging combined

with optogenetic activation of the projections in awake head-fixed mice (Fig. 5a–c and Supplementary Fig. 10). To activate PrL GABAergic projections, we injected a Cre-dependent AAV encoding ChrimsonR-tdTomato in the PrL of GAD<sup>Cre</sup> mice. An optic fiber implanted in the

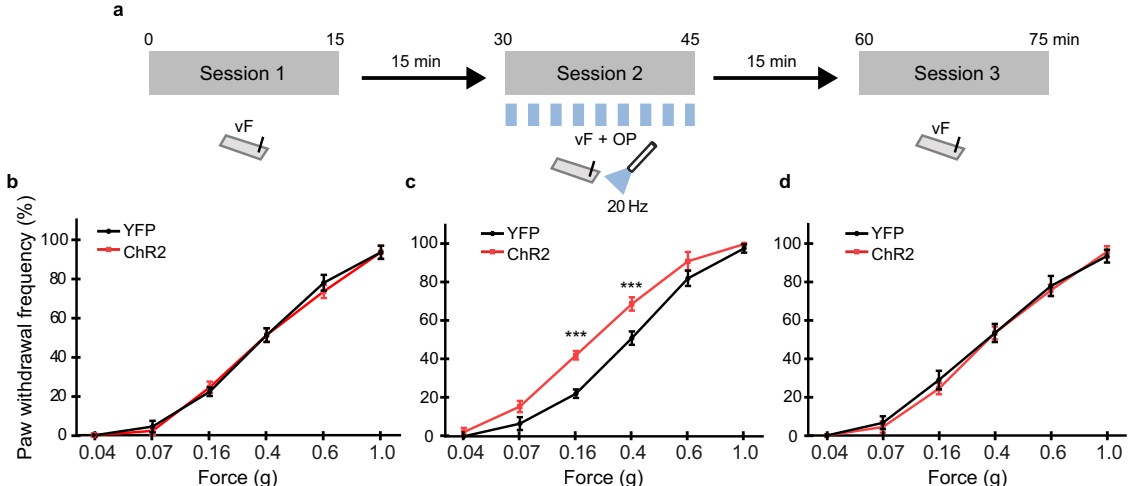

**Fig. 4 | Optogenetic activation of PrL GABAergic projections in the Cg1/2 increases the response to mechanical stimulation. a** Mice were injected with virus (AAV-DIO-ChR2-mCherry, $N = 9$ mice, AAV-DIO-YFP, $N = 9$ mice) into the PrL and an optic fiber was implanted into the Cg1/2. They were subsequently tested in a mechanical stimulation assay using von Frey filaments (session 1, pre-optical stimulation; session 2, peri-optical stimulation; session 3, post-optical stimulation) each with 6 different vF filament forces. **b** There is no difference in basal mechanical sensitivity between ChR2 and YFP groups prior to optical stimulation (all $p > 0.05$).

**c** During optical stimulation of PrL GABAergic axon terminals in the Cg1/2 (10-ms pulses at 20 Hz, 473-nm wavelength), there is a *group* v *force* interaction ($F(5,80) = 3.57$, $p = 0.0058$) and an increased sensitivity to the 0.16-g ($p < 0.0001$) and 0.4-g filaments ($p = 0.0003$). **d** Following optical stimulation, there is again no difference between the ChR2 and YFP groups in response to vF stimulation (all $p > 0.05$). All data were analyzed by RM ANOVA followed by two-sided Bonferroni corrected multiple comparisons. Lines on graphs show group means ± S.E.M., ***$p \leq 0.001$.

Cg1/2 enabled the activation of PrL GABAergic terminals expressing ChrimsonR with 633-nm red light in the Cg1/2[43] (Fig. 5b, c). We verified the reliability of optogenetic activation and excluded the potential activation via two-photon scanning (Supplementary Fig. 11). To record neuronal activity in the Cg1/2, GCaMP6s[44] was expressed under the control of the CaMKII promoter in excitatory neurons of the Cg1/2 (Fig. 5b, c). We confirmed that the expression of GCaMP6s was detected mainly in excitatory neurons[45,46] and was rarely found in GABAergic neurons of the Cg1/2 (Supplementary Fig. 12). Implantation of a gradient-index (GRIN) lens allowed imaging of the GCaMP6s signal reflecting $Ca^{2+}$ activity in Cg1/2 neurons at the single-cell resolution (Fig. 5d).

We first investigated whether activation of PrL GABAergic projections modulates ongoing activity in the absence of any stimulus. To this end, we measured mean $Ca^{2+}$ signals, peak amplitude and peak frequency of ongoing $Ca^{2+}$ signals that were recorded during baseline (session 1), optogenetic (OP)-activation (session 2) and post-activation (session 3) (Supplementary Fig. 13). In session 2, axon terminals of PrL GABAergic projections in the Cg1/2 were optogenetically activated (10-ms pulses at 20 Hz, 633-nm wavelength). The optogenetic activation of the projections did not significantly change the mean fluorescence change ratio, the peak amplitude and the peak frequency of ongoing $Ca^{2+}$ signals in the Cg1/2 of control and ChrimsonR-expressing animals in session 2. We hence conclude that PrL GABAergic projections did not affect ongoing activity in the Cg1/2.

We next investigated whether activation of PrL GABAergic projections modulates stimulus-evoked responses in the Cg1/2. We used a vF filament and applied mechanical stimulation to the plantar hind paw of mice contralateral to the Cg1/2, while simultaneously recording $Ca^{2+}$ signals through the implanted GRIN lens (Fig. 5a–d). As in the experiments above, optogenetic stimulation was restricted to session 2, whereas vF-evoked responses were measured during all three sessions (Fig. 5e). Only neurons which responded significantly to the vF stimuli (above or below 97.5% of shuffled value) during one or more of the three sessions were included in the analysis (65.0% in control mice and 66.9% in ChrimsonR mice, not significantly different, Fisher's exact test).

In all three sessions, the vF stimulus evoked increased responses in the Cg1/2 at the population level (Fig. 5f). In control animals, the

vF-evoked responses in the Cg1/2 were decreased in session 2, but not in session 3, compared to session 1 (session 1 vs. session 2, $p < 0.01$, session 1 vs. session 3, not significant, two-sided Wilcoxon signed-rank test followed by Bonferroni correction; Fig. 5f left, g). In the ChrimsonR animals, the optogenetic activation of the projections significantly enhanced the vF-evoked responses in sessions 2 and 3 compared to session 1 (session 1 vs. session 2, $p < 0.001$, session 1 vs. session 3, $p < 0.01$; Fig. 5f right, h). To compare the effect of the optogenetic activation between control and ChrimsonR mice, we determined the changes in vF-evoked responses in sessions 2 and 3 relative to session 1 (session 2, $p < 0.001$, session 3, $p < 0.01$, two-sided Mann–Whitney $U$ test followed by Bonferroni correction; Fig. 5i). The changes in the responses were significantly larger in the ChrimsonR mice compared to those in the control mice when considering the activity in all imaged layers. When restricting the analysis to the main output layer (L5) the effect remained significant but not when considering L2/3 (L5, session 2, $p < 0.001$, session 3, $p = 0.054$, two-sided Mann–Whitney $U$ test followed by Bonferroni correction; Fig. 5j, k). Thus, recruitment of PrL GABAergic terminals in the Cg1/2 enhanced the activity of the output layer.

To assess the relationship between the Cg1/2 output and the behavioral responses, we examined whether the vF-evoked $Ca^{2+}$ responses in the Cg1/2 diverged when considering trials with and without paw withdrawal (Supplementary Fig. 14a, b). The mean vF-evoked $Ca^{2+}$ responses in the withdrawal trials were larger than those in the no-withdrawal trials across all sessions in both control and ChrimsonR animals ($p < 0.05$ in all session, two-sided Wilcoxon signed-rank test followed by Bonferroni correction; Supplementary Fig. 14d–f), suggesting that the Cg1/2 responses beyond a certain threshold correlate with the paw withdrawal. We next examined whether PrL GABAergic projections modulated vF-evoked responses differentially when comparing trials with and without paw withdrawal. The activation of PrL GABAergic projections enhanced the responses compared to session 1 in the no-withdrawal trials of ChrimsonR mice (session 1 vs. session 2, $p < 0.001$, session 1 vs. session 3, $p < 0.01$, two-sided Wilcoxon signed-rank test followed by Bonferroni correction) but not those in control mice (Supplementary Fig. 14g). Furthermore, the changes in the responses relative to

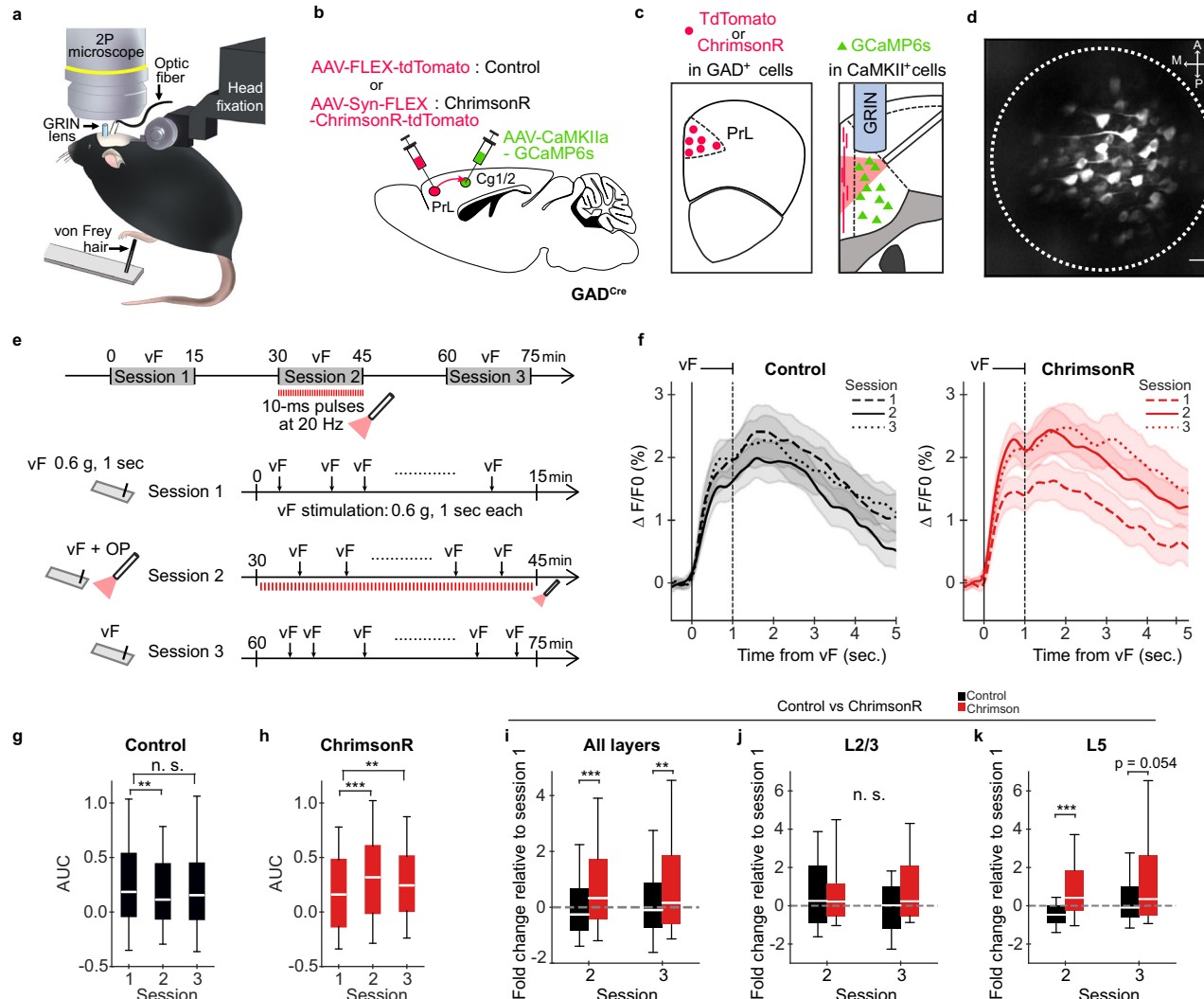

**Fig. 5 | Optogenetic activation of PrL GABAergic projections in the Cg1/2 enhances vF-evoked responses in L5. a** Schematic of the approach for Ca²⁺ imaging of Cg1/2 neurons and optogenetic stimulation of axon terminals of PrL GABAergic projection neurons in head-fixed GAD^Cre mice that received GRIN lens and optic fiber implantation. The animals were placed on a grid and received von Frey (vF) stimulation to the hind paw contralateral to the implantation site. **b** AAV-FLEX-tdTomato (control, $N = 13$ mice) or AAV-Syn-FLEX-ChrimsonR-tdTomato (ChrimsonR, $N = 15$ mice) was injected into the PrL, and AAV-CaMKIIa-GCaMP6s was injected into the Cg1/2 of GAD^Cre mice. **c** Virus-induced expression of tdTomato (control) or tdTomato-tagged ChrimsonR in GABAergic cells of the PrL. Virus-induced expression of GCaMP6s in excitatory cells, implantation of GRIN lens and illumination with red light from an optic fiber tip implanted in the Cg1/2. **d** Representative field of view through the implanted GRIN lens using the two-photon microscope showing GCaMP6 expressing neurons. Scale bar, 50 µm. **e** Schematic of the procedure used for Ca²⁺ imaging experiments, indicating stimulation details and temporal profile of vF and red-light stimulation in the three sessions. **f** Average neuronal responses to vF stimuli in Cg1/2 neurons during session 1 (stippled line), 2 (solid line) and 3 (dotted line) in control mice expressing only td-Tomato ($N = 13$ mice, $n = 307$ cells) and in ChrimsonR-expressing mice ($N = 15$ mice, $n = 397$ cells). ΔF/F0 traces were smoothed with a Gaussian filter

(s = 100 ms) for visualization purpose. Data are shown as means ± S.E.M. Black vertical line indicates the onset of the vF stimulus. Black stippled vertical line indicates the end of the stimulus. **g** Area under the curves (AUCs) of the vF-evoked responses in control ($p = 0.00872$, Friedman test, session 1 vs. 2, $p = 0.00239$, session 1 vs. 3, $p = 0.533$, DF = 306, two-sided Wilcoxon signed-rank test followed by Bonferroni correction) and **h** in ChrimsonR mice ($p = 6.05e{-}05$, Friedman test, session 1 vs. 2, $p = 1.07e{-}05$, session 1 vs. 3, $p = 0.00348$, DF = 396, two-sided Wilcoxon signed-rank test followed by Bonferroni correction). **i** Fold change of AUC for session 2 and 3 relative to session 1 in all layers (session 2, $p = 3.94e{-}07$, session 3, $p = 0.00378$, DF = 306, two-sided Mann–Whitney $U$ test followed by Bonferroni correction), **j** Fold change of AUC for session 2 and 3 relative to session 1 in L2/3 (control, $n = 38$ neurons, ChrimsonR, $n = 94$ cells, session 2, $p = 1.00$, session 3, $p = 0.132$, DF = 37, two-sided Mann–Whitney $U$ test followed by Bonferroni correction) and **k** Fold change of AUC for session 2 and 3 relative to session 1 in L5 neurons (control, $n = 127$ cells, ChrimsonR, $n = 112$ neurons, session 2, $p = 2.44e{-}06$, session 3, $p = 0.0538$, DF = 111, two-sided Mann–Whitney $U$ test followed by Bonferroni correction). Box and whisker plots in (**g**–**k**) indicate median, interquartile range and 10th to 90th percentiles of the distribution. n.s. not significant, *$p < 0.05$, **$p < 0.01$, ***$p < 0.001$, DF degree of freedom.

session 1 were greater in the ChrimsonR mice compared to those in the control mice (session 2, $p < 0.001$, session 3, $p < 0.01$, two-sided Mann–Whitney $U$ test followed by Bonferroni correction; Supplementary Fig. 14h). In the withdrawal trials, the responses were not significantly different between sessions both in control and ChrimsonR mice (Supplementary Fig. 14i). The results regarding the enhancement of Ca²⁺ responses in trials with no paw withdrawal

indicate that the activation of PrL GABAergic terminals in the Cg1/2 enhances neuronal activity when it is below the threshold for a behavioral response. In such a scenario, a proportion of sub-threshold responses would reach threshold and lead to a behavioral response. Accordingly, paw withdrawals increased during stimulation ($p < 0.01$, two-sided Wilcoxon signed-rank test followed by Bonferroni correction; Supplementary Fig. 14c).

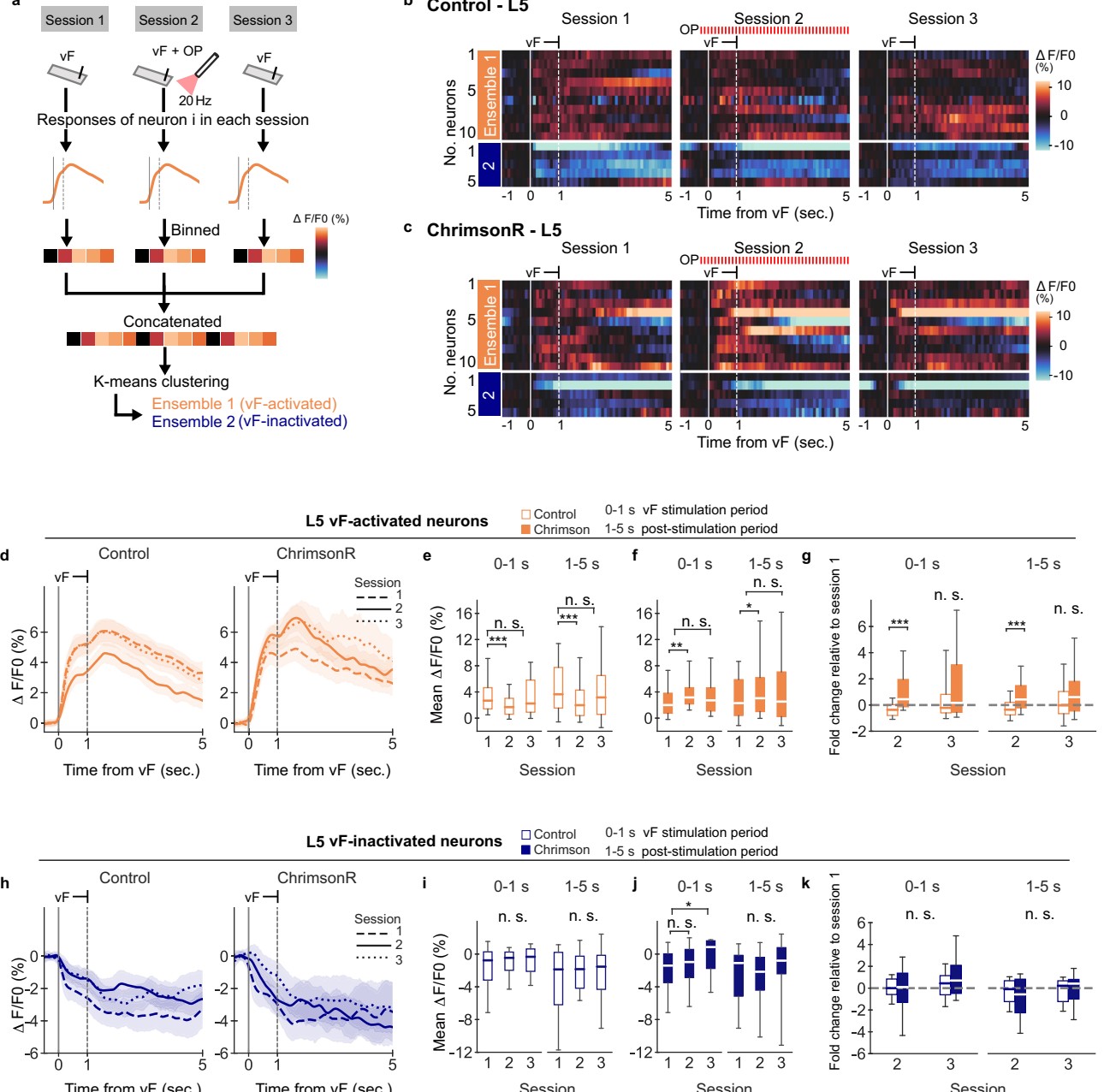

## Optogenetic activation of PrL GABAergic projections modulate activity of distinct Cg1/2 neurons

We next investigated whether Cg1/2 neurons differing in the Ca²⁺ signal in response to the vF stimulus, would also exhibit differences to the modulatory effect of PrL GABAergic neurons. To this end, neurons were divided into two groups employing unsupervised k-means clustering based on the mean Ca²⁺ activity around vF stimuli (Fig. 6a). One group (referred to as the "vF-activated" neurons) displayed increased activity and the second group ("vF-inactivated" neurons) exhibited decreased activity after onset of the vF stimuli (Supplementary Fig. 15a, b). The proportion of vF-activated and inactivated neurons were not significantly different in control and ChrimsonR mice (control, vF-activated 61.6%, vF-inactivated 38.4%; ChrimsonR, vF-activated 64.0%, vF-inactivated 36.0%, Fisher's exact test; Supplementary Fig. 15c).

To dissect putative acute vs. long-lasting modulatory effects of PrL GABAergic projections on Ca²⁺ responses during vF stimulation, we analyzed vF-evoked responses in two-time windows after vF stimulus

onset, namely from 0 to 1 s (stimulation period) and from 1 to 5 s (post-stimulation period). Consistent with the previous analysis (Fig. 5), optogenetic activation of PrL axons in session 2 affected L5 vF-activated neurons but not L2/3 vF-activated neurons (Fig. 6 and Supplementary Fig. 16). Activation of PrL GABAergic projections in session 2 significantly enhanced the response of L5 vF-activated neurons to the vF stimulus in ChrimsonR mice ($p < 0.01$, two-sided Wilcoxon signed-rank test followed by Bonferroni correction), which was not the case in control mice (Fig. 6b–f). In fact, in control mice the response of L5 vF-activated neurons was decreased ($p < 0.001$, two-sided Wilcoxon signed-rank test followed by Bonferroni correction; Fig. 6e). Notably, in both control and ChrimsonR mice the response in session 3 (without optogenetic stimulation) was not significantly changed (Fig. 6e, f). Activation of PrL GABAergic neuron axons resulted in significantly larger changes in responses of L5 vF-activated neurons in ChrimsonR mice compared to control mice during both the 0–1 s vF stimulation period and the 1–5 s post vF stimulation period in session 2 (both $p < 0.001$, two-sided Mann–Whitney $U$ test followed by Bonferroni

**Fig. 6 | Optogenetic activation of PrL GABAergic projections differentially modulates responses of Cg1/2 L5 neurons during and after vF stimulation.**
**a** Schematic of k-means clustering, illustrating data transformation of vF-evoked responses. Example of a neuron which became activated upon vF stimulation. Neuronal responses to vF stimulation during session 1–3 are binned and then concatenated before k-means clustering is performed. **b** Heatmap indicating mean vF-evoked responses from representative L5 neurons of control and **c** Chrimson mice in each session. Data are assigned to two ensembles based on k-mean clustering. White vertical line and stippled line indicate onset and end of the vF stimulus. The responses were sorted based on the maximum value during vF stimulation in session 1. **d** Average neuronal responses to vF stimuli of L5 vF-activated neurons during session 1 (stippled line), 2 (solid line) and 3 (dotted line) in control ($n = 81$ cells) and in ChrimsonR mice ($n = 81$ neurons). Gray vertical and stippled lines indicate onset and end of vF stimulus, respectively. **e** Mean $\Delta F/F0$ of L5 vF-activated neurons during (0–1 s) and post (1–5 s) vF stimulation in control (open box, during vF stimulation in session 1 vs. 2, $p = 1.28e{-06}$, session 1 vs. 3, $p = 1.00$, post vF stimulation in session 1 vs. 2, $p = 5.04e{-05}$, session 1 vs. 3, $p = 1.00$, DF = 80, two-sided Wilcoxon signed-rank test followed by Bonferroni correction) and **f** in ChrimsonR-expressing mice (filled box, during vF stimulation in session 1 vs. 2, $p = 0.00167$, session 1 vs. 3, $p = 0.179$, post vF stimulation in session 1 vs. 2, $p = 0.0449$, session 1 vs. 3, $p = 0.0826$, DF = 80, two-sided Wilcoxon signed-rank test followed by Bonferroni correction). **g** Fold change of $\Delta F/F0$ means for session 2 and 3 relative to session 1 during and post vF stimulation in control (open box) and

ChrimsonR mice (filled box, during vF stimulation in session 2, $p = 5.58e{-10}$, session 3, $p = 0.238$, post vF stimulation in session 2, $p = 1.24e{-05}$, session 3, $p = 0.139$, DF = 80, two-sided Mann–Whitney $U$ test followed by Bonferroni correction).
**h** Average neuronal responses to vF stimuli of L5 vF-inactivated neurons during session 1 (stippled line), 2 (solid line) and 3 (dotted line) in control ($n = 46$ neurons) and in ChrimsonR mice ($n = 31$ neurons). Gray vertical and stippled vertical lines indicate onset and end of vF stimulus, respectively. **i** Mean $\Delta F/F0$ of L5 vF-inactivated neurons during (0–1 s) and post (1–5 s) vF stimulation in control (open box, during vF stimulation in session 1 vs. 2, $p = 1.00$, session 1 vs. 3, $p = 0.438$, post vF stimulation in session 1 vs. 2, $p = 1.00$, session 1 vs. 3, $p = 0.668$, DF = 45, two-sided Wilcoxon signed-rank test followed by Bonferroni correction) and **j** in ChrimsonR-expressing mice (filled box, during vF stimulation in session 1 vs. 2, $p = 1.00$, session 1 vs. 3, $p = 0.0243$, post vF stimulation in session 1 vs. 2, $p = 1.00$, session 1 vs. 3, $p = 1.00$, DF = 30, two-sided Wilcoxon signed-rank test followed by Bonferroni correction). **k** Fold change of $\Delta F/F0$ means for session 2 and 3 relative to session 1 during and post vF stimulation in control (open box) and ChrimsonR mice (filled box, during vF stimulation in session 2, $p = 1.00$, session 3, $p = 0.820$, control vs. ChrimsonR, post vF stimulation in session 2, $p = 1.00$, session 3, $p = 1.00$, DF = 30, two-sided Mann–Whitney $U$ test followed by Bonferroni correction). **d, h** $\Delta F/F0$ traces were smoothed with a Gaussian filter ($s = 100$ ms) for visualization purpose. Data are shown as means ± S.E.M. **e–g**, **i–k** Box and whisker plots indicate median, interquartile range and 10th to 90th percentiles of the distribution. n.s. not significant, *$p < 0.05$, **$p < 0.01$, ***$p < 0.001$, DF degree of freedom.

---

correction, Fig. 6g). Thus, the modulatory effect outlasted the vF stimulation period.

The modulatory effect of optogenetic PrL GABAergic projection activation on responses of vF-activated neurons was not seen in L5 vF-inactivated neurons in session 2 (Fig. 6h–k). Only in session 3, the response of the L2/3 and L5 vF-inactivated neurons in ChrimsonR mice was increased compared to session 1 during vF stimulation (0–1 s), suggesting that there was a reduced suppression in vF-inactivated neurons (L2/3, $p < 0.01$, L5, $p < 0.05$, two-sided Wilcoxon signed-rank test followed by Bonferroni correction; Fig. 6j and Supplementary Fig. 16i).

A direct comparison between panels f and j in Fig. 6 that illustrate the changes in vF-activated and vF-inactivated neurons during and post-optogenetic activation in ChrimsonR mice suggests that different mechanisms may account for the overall modulatory effect of PrL GABAergic projection activation in session 2 and 3. Thus, optogenetic activation of PrL GABAergic terminals in the Cg1/2 affects vF-activated, but not vF-inactivated neurons during optogenetic activation (i.e., session 2). However, in the session after optogenetic activation (i.e., session 3) there is an increased response during the 0–1 s vF stimulation period in vF-inactivated neurons but no effect in vF-activated neurons.

A previous study demonstrated that, compared to weaker vF stimuli, stronger vF stimuli increased the proportion of responsive L5 pyramidal neurons in the Cg1/2[47]. Since the mice were more sensitive to mechanical stimuli during the activation of PrL GABAergic projections (Fig. 4), we considered the possibility that the activation of PrL GABAergic projections might increase the number of responsive Cg1/2 neurons. This was indeed the case: We focused on vF-activated neurons and determined the proportion of neurons that responded to vF stimuli in each session (Supplementary Fig. 17). In control mice, the proportion of responding neurons was decreased in sessions 2 and 3 compared to session 1, revealing that vF-activated neurons in the Cg1/2 displayed attenuated responsiveness upon repeated vF stimulation ($p < 0.001$, $\chi^2$ test, session 1 vs. session 2, $p < 0.001$, session 1 vs. session 3, $p < 0.01$, Fisher's exact test followed by Bonferroni correction; Supplementary Fig. 17a). In contrast, in ChrimsonR mice, the proportion of neurons was significantly higher in session 2, and returned to the baseline proportion in session 3 ($p < 0.001$, $\chi^2$ test, session 1 vs. session 2, $p < 0.001$, session 1 vs. session 3, n.s., Fisher's exact test followed by Bonferroni correction; Supplementary Fig. 17b). These results suggest that activation of PrL GABAergic projections in the Cg1/2 increased the number of neurons responsive to vF stimulation.

Together, these results support the notion that activation of PrL GABAergic projections in the Cg1/2 changes the processing of vF stimuli by regulating both the activity within given neuronal ensembles and the ensemble size that contributes to sensory-response processing.

## Discussion

Here we demonstrate that two areas of the mPFC are connected via GABAergic projection neurons. Specifically, we characterized the projection from the PrL to the Cg1/2 at multiple levels: In vitro electrophysiological recordings revealed the identity of target cells, that is L1 INs. Based on these results together with those obtained from the $Ca^{2+}$ imaging studies in vivo, we propose a mechanism according to which GABAergic projections affect the activity of L5 output cells in the Cg1/2, thereby revealing how interareal connectivity between the PrL and the Cg1/2 shapes top-down signaling and modifies behavior.

Our histological and electrophysiological experiments showed that axons of PrL GABAergic projection neurons are by and large confined to the most superficial cortical layer, i.e., L1. In the superficial layers, PrL GABAergic projection neurons innervate specifically GABAergic INs in L1. Notably, the GABAergic projection from the PrL to the Cg1/2 is reciprocated by a GABAergic projection in the opposite direction, that is from the Cg1/2 to the PrL. Importantly the connectivity pattern is identical in that L1 GABAergic INs appear to be the exclusive target of this projection. The fact that both projections inhibit local INs in the target area is reminiscent, but not identical, to the connectivity pattern of other GABAergic projection neurons in the basal forebrain. For instance, septal GABAergic projection neurons to the medial entorhinal cortex also inhibit INs selectively, however, the target cells reside in L2[21]. Similarly, hippocampal GABAergic projection neurons target INs in L2 of the medial entorhinal cortex[29]. Cortico-cortical GABAergic projections, as shown here, seem to adhere to the rule according to which the target cells are INs, however, these are special in that they are L1 INs.

L1 neurons have lately received increasing attention. Their presence in L1 of multiple brain regions, including somatosensory[48,49], visual[26,50,51], auditory[31], prefrontal cortex[52] in rodents and human[53], together with functional evidence that their recruitment shapes the activity of local circuits[52,54–56] strongly suggest that L1 INs are crucial members of the cortical canonical circuit. Based on in vitro electrophysiological work combined with anatomical reconstruction of the recorded cells, previous studies distinguished two major

morphological types of L1 INs[50,57], namely NGFs and SBCs. Although much remains to be investigated regarding the connectivity of L1 INs and the effects of their recruitment on L5 neurons[56], there is evidence that the differential connectivity of NGFs and SBCs with neurons in the layers underneath accounts for the opposite effect in L5 output neurons[30,54,55]. Thus, recruitment of NGFs would induce an overall inhibitory effect in L5 neurons whereas recruitment of SBCs would support disinhibition in L5 neurons. GABAergic projection neurons in turn, as shown in this study, appear to be gatekeepers that control neuronal activity in the target area by inhibiting selectively L1 INs. Thus, GABAergic projection neurons put a break on L1 NGFs and SBCs, thereby reducing or possibly even inversing their inhibitory and dis-inhibitory effect on L5 pyramidal neurons, respectively. Since opto-genetic activation of PrL GABAergic projections resulted in enhanced activity of L5 Cg1/2 neurons, we infer that the modulatory effect involved NGFs in L1. A putative recruitment of SBCs under the stimu-lation conditions was not detectable, at least when considering L5 pyramidal cells as a readout (recruitment of SBCs would entail the opposite effect). One possible explanation is that NGFs may have higher spontaneous activity than SBCs. However, we did not detect differences in the mean frequency or the mean amplitude of sponta-neous EPSC (sEPSC) in pSBCs and pNGFs (Supplementary Fig. 18). Nevertheless, given the changes induced by the stimulation of GABAergic projections both at the network level (Ca²⁺ imaging) and at the behavioral level, we infer the involvement of NGFs. This may reflect a broader control of NGFs compared to SBCs under the conditions tested here. The two cell types differ in that the former arborize hor-izontally whereas the latter vertically[30]. This raises the question as to when the two distinct types of L1 INs are normally recruited under physiological conditions, not only in the areas studied here, but also in other cortical areas. At the moment it is unclear when the GABAergic projection neurons investigated here counterbalance/modulate the activity of cortico-cortical and thalamo-cortical glutamatergic projec-tions or of cholinergic basal forebrain-derived projections that all impinge on L1 INs[58]. In addition, L1 INs are directly activated by basal forebrain-derived cholinergic input[59-62]. At least for NGFs it was demonstrated that their recruitment in visual cortex was arousal-dependent[51]. To answer the question regarding the state-dependent recruitment of L1 INs becomes a dauting task if one considers that the two L1 IN cell types can be further subdivided if molecular markers are considered[49,63].

At the behavioral level, activation of PrL GABAergic projections to the Cg1/2 affected responses both in the vF test and the visual atten-tion task. In the vF test, the increase in paw withdrawal frequency corresponded to an enhancement of vF-evoked responses in the Cg1/2. When separating the vF-evoked responses on the basis of the beha-vioral consequences (i.e., paw withdrawal vs. no paw withdrawal), the vF-evoked Ca²⁺ responses in the presence of paw withdrawal were larger than those in the absence of paw withdrawal. These results indicate that there is a threshold for Cg1/2 responses above which paw withdrawal is induced. Accordingly, the likelihood of reaching threshold for paw withdrawal was increased by the activation of PrL GABAergic projections. Consistent with this, the anterior cingulate cortex in the human brain has been proposed to contribute to sensory driven decision making by adjusting the distance to the threshold for response[64]. Interestingly, we found that the activation of the PrL GABAergic projections enhanced the vF-evoked response, especially in the absence of withdrawal. Such an enhancement below threshold can reduce the span to the response threshold. Thus, PrL GABAergic pro-jections to the Cg1/2 might increase the likelihood of paw withdrawal by enhancing the response of Cg1/2 L5 PNs. This hypothesis is further corroborated by the experiments involving the visual attention task. Here the activation of PrL GABAergic projections in the Cg1/2 increased premature responses which are an index of impulsiveness. A similar selective increase in premature responses was observed in a

previous study where Cg1/2 L5 PNs where chemogentically activated[1]. Taken together, we conclude that PrL GABAergic projections to the Cg1/2 increase the tendency to trigger motor reactions via Cg1/2 L5 PNs activation. Furthermore, as this occurs whether the sensory sti-mulus is visual or mechanical, rather than being sensory specific this effect is likely to reflect a global shift in responsiveness.

Interestingly, both at the network level (Ca²⁺ imaging) and at the behavioral level (vF test with capsaicin injection), the enhanced response induced by the optogenetic activation persists in the post-optogenetic stimulation session (session 3 in Fig. 5 and Supplementary Fig. 9). Given the vF stimulation protocols used in these experiments, i.e., repeated application of the vF stimulus and application of the vF stimulus under acute pain, the prolonged enhancement of the vF-evoked responses may well reflect synaptic plasticity induced during optogenetic activation. This would be in agreement with experimental data supporting the notion that synaptic plasticity in the Cg1/2 in rodents contributes to chronic pain[65-69]. Whilst there is ample evidence that anterior cingulate cortex is involved in cortical processing underlying both acute and chronic pain[69,70], it is still largely unknown which mechanisms trigger the transition from acute to chronic pain and how synaptic plasticity and the thereby induced altered proces-sing contributes therein. It is tempting to speculate that top-down signaling mediated by PrL GABAergic projection neurons contributes to this transition, and that the specific inactivation of PrL projection neurons may reduce the development of chronic pain.

The data that we presented and discussed here are important for the following reasons: Firstly, we identified GABAergic projection neurons that connect the PrL and the Cg1/2 and modulate neuronal activity in the target area thereby affecting top-down signaling. Their recruitment regulates the output in the Cg1/2 in the opposite fashion compared to the excitatory projection that connect the two areas. Secondly, we propose that the underlying mechanism involves inhi-bition of GABAergic INs in L1 whereby they modulate neuronal activity in the target area. Thirdly, we propose cortico-cortical modulatory control is very likely to play a role in yet other brain areas. It is safe to make this assumption for two reasons: there is anatomical evidence for GABAergic projection neurons whose axons travel in L1 connecting different brain areas. Furthermore, L1 INs are found across the entire neocortex. They were shown before to be in the optimal position to control local neuronal networks and the behaviors associated therewith[56]. In sum, we propose a scenario in which communication across cortical areas is realized by linking cortico-cortical GABAergic projection neurons to canonical circuits via L1 INs thereby governing integrative processes in global cortical computation (Supplemen-tary Fig. 19).

## Methods
### Animals
Experiments were performed on naive male mice described before GAD^cre, *GAD67*^EGFP71, *SOM*^Cre29, *PV*^Cre72, *VIP*^Cre73, *5HT*^EGFP74. The mice were maintained on a C57BL/6N background and were 8–14 weeks old at the start of the experiment. Mice were maintained on a 12-h light/dark schedule at $21 \pm 2\,°C$ and $50\% \pm 10\%$ humidity, and the experiments were performed during the light phase of the cycle. During the operant task mice were food restricted and maintained at >85% of their free-food body weight. Water was always available ad libitum in the home cage. All experiments were approved by the Regierungspraesidium Karlsruhe (AZ 35-9185.81/G-119/14, 35-9185.81/G-157/16, 35-9185.81/G-61/18, 35-9185.81/G-131/19) in compliance with the European guidelines for the care and use of laboratory animals.

### Virus injection and implantation
Mice were mounted in a stereotactic apparatus under 0.5–2% iso-flurane anesthesia during surgery. Craniotomies were made above the injection sites. From glass pipettes, viruses were injected using the

coordinates from bregma. Detailed information about the injection coordinates and injected viruses can be found in Supplementary Tables 2–4.

For anterograde tracing, the mice were injected with AAV1-CAG-Flox-eGFP-WPRE-bGH (50 nl, Penn Vector Core) and AAV1-CaMKIIa-hChR2(h134a)-mCherry (50 nl, Addgene) into the PrL or the Cg1/2. For retrograde tracing, Cholera Toxin subunit B (Alexa Fluor 555 Conjugate, 100 nl, Thermo Fisher Scientific) was injected into the Cg1/2 or the PrL. The pipette was kept in the brain for 8 min before retraction to avoid diffusion of the virus. After the injection, the scalp was sutured. Injection of carprofen was given as a post-surgery analgesic. The mice were allowed to recover for 4 weeks before perfusion.

For electrophysiological studies, the mice were injected with AAV1-double floxed-hChR2(H134R)-mCherry[75] (AAV-DIO-ChR2-mCherry, Addgene, 100 nl) into the PrL. After the injection, the scalp was sutured. Injection of carprofen was given as a post-surgery analgesic. After surgery, mice were housed for 3 weeks before performing experiments.

For the operant task, 50 nl of AAV-DIO-ChR2-mCherry or AAV1-Ef1a-DIO EYFP (AAV-DIO-YFP, Addgene) was injected into the PrL. Two screws were secured onto the skull, and a 200-µm diameter optic fiber (0.22 NA, CFML22U, Thorlabs) was then lowered into the Cg1/2 (coordinates from bregma: AP +1.2, ML +1.0, DV −1.2) at an angle of 40 degrees toward the midline (to avoid the optic fiber damaging the superficial layers of the Cg1/2). The optic fiber was anchored to the screws and skull using dental cement. Opaque nail varnish was added to the dental cement to prevent the cement becoming illuminated during laser stimulation. Mice were allowed >10 days for recovery prior to being placed on food control.

For the von Frey test, AAV-DIO-ChR2-mCherry (100 nl) or AAV-DIO-YFP (100 nl) were injected into the PrL. A 200-µm diameter optic fiber (CFML22U, Thorlabs) was then lowered into the Cg1/2 (coordinates from bregma: AP +1.2, ML +1.0, DV −1.2) at an angle of 40 degrees toward the midline.

For in vivo $Ca^{2+}$ imaging, the mice were injected with AAV5-Syn-FLEX-rc[ChrimsonR-tdTomato] (150 nl, Addgene) or AAV1-FLEX-tdTomato (150 nl, Addgene) into the PrL and AAV9-CamKII.GCaMP6s.WPRE.SV40 (200 nl, Addgene) into the Cg1/2. Subsequently, the optic fiber (CFML22U, Thorlabs) was implanted (AP +1.20 mm, ML ±1.9 mm, DV −0.9 mm, at an angle of 60 degrees toward the midline) and fixed to the skull using light-curable dental cement (Gradia Direct Flo BW, GC Europe, 002358). In addition, a GRIN microendoscope (GRIN lens, 0.5 × 6.48 mm, GRINTECH) was implanted into the Cg1/2. The GRIN lens was implanted lateral to the sinus to avoid damaging it, which also placed the lens lateral to L1 of the ACC. This also ensures the intactness of L1. A small blade was used to make a small incision for implantation. The GRIN lens was lowered into the brain (DV −0.6 mm) with a lens holder and fixed to the skull together with the implanted optic fiber and a headpost (Luigs und Neumann) using cement. Carprofen was administered as a post-surgery analgesic. The mice were allowed to recover for 4 weeks after surgery before starting habituation for in vivo imaging.

## Immunohistochemistry
Mice were transcardially perfused with 4% paraformaldehyde (PFA) under deep anesthesia (ketamine, 20%, 50 mg/ml, xylazine, 8%, 20 mg/ml). The brains were dissected and fixed in 4% PFA at 4 °C overnight. Coronal sections were cut at 50 or 100 µm thicknesses with a vibratome (Leica VT1000S) and washed with phosphate-buffered saline (PBS) three times for 5 min. Floating sections were permeabilized and blocked for 2 h with PBS containing 5% BSA and 0.1% Triton X-100. The incubation of the sections with primary antibodies was performed for 24 h (18 h for cFos) at 4 °C. The sections were subsequently washed with PBS three times for 10 min and incubated for 2 h with secondary antibodies and 4,6-diamidino-2-phenylindole (DAPI, Invitrogen, D1306, final dilution 1:10,000) for 30 min at room temperature. The

sections were then washed with PBS three times for 10 min and mounted on glass slides. The sections were sealed with Mowiol 40-88 (Sigma-Aldrich, 324590) and covered with coverslips. Confocal images of the sections were taken using a Zeiss LSM 700 confocal microscope (Carl Zeiss). As primary antibodies we used chicken anti-EGFP (1:1000, Invitrogen, A10262), rabbit anti-DsRed (1:1000, Takara Bio, 632496), rabbit anti-cFos (1:2000 abcam ab190289). As secondary antibodies we used Alexa 488-conjugated anti-chicken IgY, Alexa 647-conjugated anti-rabbit IgG (1:1000, Invitrogen) and Cy3 AffiniPure Donkey Anti-Rabbit (1:1000 Jackson Immunoresearch Cat# 711-165-152) Cy5 AffiniPure Donkey Anti-Rabbit (1:1000 Jackson Immunoresearch Cat# 711-175-152).

For quantification of laminar distribution of axons, the axons were manually traced and skeletonized using Fiji ImageJ (NIH, v 2.0.0). The length of the skeletonized axons was measured with ImageJ plugin AnalyzeSkeleton[76].

For detection of cell bodies, retrogradely labeled (CTB+ and GFP+) and virally labeled cells (GCaMP6s+, tdTomato+) in the images were counted using Fiji ImageJ (v 2.0.0).

## Electrophysiological recordings
For in vitro patch-clamp recordings, mice were subjected to deep anesthesia using isoflurane and then transcardially perfused with approximately 20 ml of NMDG aCSF containing the following (in mM) 92 NMDG, 2.5 KCl, 1.25 $NaH_2PO_4$, 30 $NaHCO_3$, 20 HEPES, 25 glucose, 2 thiourea, 5 Na-ascorbate, 3 Na-pyruvate, 0.5 $CaCl_2$ and 10 $MgCl_2$, and oxygenated with carbogen gas (95% $O_2$ and 5% $CO_2$, pH 7.3–7.4). Following decapitation, the brains were removed and 400 µm thick coronal sections were obtained on a slicer in oxygenated NMDG aCSF. Slices were initially recovered in oxygenated NMDG aCSF at 34 ± 1 °C for 11 min followed by incubation in HEPES holding aCSF containing the following (in mM) 92 NaCl, 2.5 KCl, 1.25 $NaH_2PO_4$, 30 $NaHCO_3$, 20 HEPES, 25 glucose, 2 thiourea, 5 Na-ascorbate, 3 Na-pyruvate, 2 $CaCl_2$ and 2 $MgCl_2$, and oxygenated at 23 ± 1 °C until they were used. During experiments, slices were placed in a recording chamber and superfused with oxygenated aCSF containing (in mM): 125 NaCl, 25 $NaHCO_3$, 1.25 $NaH_2PO_4$, 2.5 KCl, 25 glucose, 2 $CaCl_2$ and 1 $MgCl_2$ at 31 ± 1 °C unless elsewhere noted.

Cells and mCherry-labeled projections were visualized using infrared-differential interference contrast and epifluorescence microscopy, respectively (Olympus BX51WI and U-MWIG3 coupled with a CCD camera, SciCam Pro, Scientifica). Recording and stimulating electrodes as well as puffing pipettes (3–5 MΩ) were pulled from borosilicate glass with a filament (O.D. 1.5 mm, I.D. 0.86 mm; GB150F-8P, Science Products, Germany). For testing the connectivity in L1 and L2/3 neurons, whole-cells recordings were made with a recording electrode filled with high Cl⁻ internal solution ($E_{GABA}$ = -2 mV) containing the following (in mM): 127.5 KCl, 11 EGTA, 10 HEPES, 1 $CaCl_2$, 2 $MgCl_2$, 2 MgATP and 0.5% biocytin. Neuronal classification was done based on distinct electrophysiological properties and on the location of the neurons in a given neocortical layer[21,30]. L1 INs were classified as putative neurogliaform cells (pNGF) or putative single-bouquet cells (pSBC) based on first spike latency and first afterhyperpolarization (AHP) latency. The two types of L1 INs differed also significantly with respect to their depolarizing hump amplitude (Supplementary Table 1 and Supplementary Fig. 7). The input resistance was estimated from the ratio of median voltage responses to current injection (1 s, ±30 pA, 10 pA increments). The membrane time constant was estimated from the time constant of the single exponential fit (fit range, 0.1 s) to voltage relaxation in response to current injection (1 s, −50 pA). The sag ratio was defined as the median of steady-state voltage response (0.3 s from the end of current injection, −50 pA) divided by the minimum voltage response in 0.3 s after the start of current injection. The first spike latency was determined by the difference in time between the peak of the first action potential (AP) and the start of current injection.

The amplitude of the depolarizing hump, for pSBCs, was calculated as the difference in potential between the threshold of first AP and the potential 0.3 s after the start of current injection. As pNGFs have a delayed spike, the amplitude of depolarizing hump was calculated as the difference between the maximal potential in the first 0.3 s and the membrane potential at 0.3 s after the start of current injection[49]. The AP amplitude was defined as the difference in potential between the peak amplitude of AP and the AP threshold. The AP threshold was defined as the nearest membrane voltage when the change of voltage over time was higher than 50 V/s. The first AHP latency was defined as the difference in time between AP threshold and the minimum of AHP. L2/3 INs were classified as putative somatostatin-expressing cells (pSOM+) when displaying a prominent hyperpolarizing sag, low-threshold firing and spike adaptation, as putative parvalbumin-expressing cells (pPV+) when exhibiting non-adapting firing of action potentials at >80 Hz, and putative 5-HT$_3$A receptor-expressing cells (p5HT$_3$R+) when presenting an irregular firing pattern (Supplementary Fig. 20). Pyramidal neurons (PN) were identified based on large and triangular somata, and an accommodating firing pattern. Note that p5HT$_3$R+ cells in the current study indicate non-NGF and non-SBC 5HT$_3$R-positive neurons in L2/3. However, most of the pSBCs (13 out of 15 cells), but not pNGFs (0 out of 15 cells), were sensitive to the 5HT$_3$R agonist mCPBG (50 μM, 0.1-s puffs, 0.5 bar, Supplementary Fig. 21). For local puffing, pipettes were filled with mCPBG-containing aCSF and were placed ~20 μm away from the recorded cell.

To measure the optically evoked postsynaptic currents (PSCs), mCherry-expressing axonal fibers were stimulated using brief laser pulses (473 nm, 5 ms, 25 mW, CL473-025-O, CrystaLaser) guided by an optical fiber (M138L02, Thorlabs), and cells were voltage clamped at −70 mV. In a subset of experiments, SR95531 (SR, 1 μM, BN0507, Biotrend) was used to test GABAergic responses.

To test for specificity of ChrimsonR activation, we performed experiments at room temperature (~20 °C) at two wave lengths of the light source, namely either at 920 nm using a two-photon microscope (Leica TCS SP5 MP) equipped with a 20X objective (HCX APO L 20X/1.00 W) or at 633 nm using another laser source (5 ms, ~4 mW, DL633-050-O, CrystaLaser). TdTomato-expressing GABAergic neurons in PrL were visually selected under infrared Dodt gradient contrast (IR-DGC) optics with epifluorescence. Cell-attached recordings were made to detect suprathreshold activity and whole-cell recordings were performed to validate the sensitivity of ChrimsonR to the different light sources (633 nm and 920 nm). The holding potential was set to −80 mV to avoid escaped action currents.

To stimulate L1 inputs, a stimulating electrode filled with aCSF was placed within L1. Paired-pulses (100 μs) at 20 Hz at near-threshold intensities were delivered every 5 s using a stimulus isolator (Model 2100, A-M SYSTEMS). To measure the electrically evoked responses, cell-attached and whole-cell recordings were made using a low Cl⁻ internal solution ($E_{GABA}$ = −92 mV) containing the following (in mM): 130 K-gluconate, 10 Na-gluconate, 10 HEPES, 10 phosphocreatine, 4 NaCl, 4 MgATP, 0.3 GTP. Putative antidromically evoked spiking activities (short latency and low jitter of spike timing) were excluded from analysis. To estimate the net effect of PrL GABAergic projections on L5 PNs, compound PSCs (including EPSC-IPSC sequences and putative pure EPSCs) evoked by electrical stimulation at L1 were recorded at near −50 mV, a holding potential between the equilibrium potential of glutamate and GABA receptors. The putative pure EPSCs were recorded at −92 mV, the equilibrium potential of ionotropic GABA receptors ($E_{GABA}$). To isolate the feedforward IPSC, the initial slope of compound PSC was scaled to the initial slope of pure EPSC recorded at −92 mV and subtracted by the pure EPSC recorded at −92 mV[77]. To minimize the IPSCs derived from the antidromically activated GABAergic neurons, the stimulating electrode was placed at least 800 μm away from the recording cell, and the stimulating intensity was tuned to a level which not always elicits EPSC-IPSC

sequences recorded at near −50 mV. Because antidromic stimulation can reliably and temporally precisely activate neurons, data were excluded if the incidence of apparent outward IPSCs recorded at near −50 mV was 100% or if the latency between the onset of the feedforward IPSC and of the EPSC was shorter than 1 ms. In our dataset, the average latency was 3.03 ± 0.45 ms ($n$ = 12 cells, $N$ = 4 mice).

To measure sEPSC, L1 INs were clamped at −92 mV ($E_{GABA}$). Stable recording periods (200 s) were used for event detection. Spontaneous events from 15% of total recording duration were first automatically detected by Mini Analysis (6.0.3, Synaptosoft) followed by manual inspection and adjustments of parameters until the summation of false positive and false negative events from automatic detection were less than 10% of total events detected manually. The parameters were then used for the rest of the recording period.

All recordings were made using an EPC 10 amplifier (HEKA, Germany). Pipette capacitance was maximally compensated. Series resistance (≤25 MΩ) was fully compensated in current clamp mode (correction 100% with a 10-μs lag). Data were included only if the series resistance changed less than 20%. Liquid junction potentials were not corrected. Stimulus delivery and data acquisition was performed using Patchmaster software (v 2 × 90, HEKA, Germany). Signals were filtered at 3 kHz and the sampling rate was 20 kHz. Data analysis was performed using Clampfit (10.7, Molecular Devices, USA) or custom written scripts in Matlab (2020a, MathWorks, USA). Data are presented as mean ± S.E.M., $n$ indicates the number of cells, and $N$ indicates the number of animals. Statistical significance was tested using GraphPad Prism 5.0.

### Biocytin staining and morphological reconstruction
Neurons were filled with biocytin (~0.5%) and outside-out patch excisions were performed to close membrane. Slices were then fixed overnight with 4% paraformaldehyde (PFA) in phosphate-buffered saline (PBS, 0.1 M). After washing with PBS, slices were incubated in 0.3% Triton X-100 in PBS (PBST) for 30 min, followed by incubation in 10% normal goat serum (NGS) in PBS for 2 h. Slices were subsequently incubated overnight in streptavidin-conjugated Alexa 546 (1:500) and 5% NGS in PBST. After washing with PBS, slices were embedded in a mounting medium (H-1200, Vectashield).

For reconstruction of biocytin-labeled cells, confocal images of the sections (voxel size, 156–313 nm in the $x$−$y$ plane, $z$-step, 1 μm) were taken using a Zeiss LSM 700 confocal microscope (Carl Zeiss). Image stacks were imported into Neuromantic 1.6.3 software[78], and 3D morphological reconstructions were made with assistance of semi-auto function. To quantify the axonal density along the horizontal and vertical axes, we centered the soma, and counted the number of intersections made by the axon with lines running vertical and parallel to the cortical layers[79].

### Operant task
The training protocol was based on studies assessing sustained attention in rats[38,39]. Training and testing were conducted in Med Associates mouse operant chambers (ENV-307W-CT, Med Associates, Fairfax, VT, USA) located inside individual sound and light attenuating cubicles (ENV-018V). Each chamber was equipped with a light on one wall (the "house light") and a lever and pellet dispenser receptacle on the opposing wall. An optic fiber patch cable (200 μm, 0.2NA, M86L005, Thorlabs) was attached to the top of the cubicle and entered the operant chamber through a custom lid. Mice were connected to the optic fiber at the start of every session in order to habituate them to the presence of the cable. The house light was programed to have four brightness levels: "off" (0 volts input), "dim" (11 volts), "bright" (17 volts) and "full" (24 volts). Prior to the start of training, mice were food restricted to 85%–90% of their free-food weight. Mice were initially trained on a Fixed Ratio (FR) 1 schedule of reinforcement. A trial started with the house light at "full" intensity,

pressing the lever once led to the simultaneous delivery of a reward pellet and extinction of the house light. The next trial began after a 10 s delay period during which pressing the lever had no programmed consequence. Once mice earned 60 pellets within one session, they were trained in the low demand version of the visual attention task. In this phase of training, each trial started with 10–20 s (duration picked randomly trial-to-trial) of darkness followed by the anticipatory phase where the house light was "dim" for 5–8 s (duration picked randomly trial-to-trial). The house light was then increased to "full" intensity for up to 20 s during which time pressing on the lever led to the delivery of a reward pellet, extinction of the house light, and end of the trial. Lever presses at any point other than the stimulus light at full intensity did not cause a reward to be delivered and led to the termination of the current trial and start of the next trial (Fig. 3b). Daily sessions ended after completing 50 correct trials or after 60 min. Once mice completed 50 correct trials within a session, the duration of the "full" intensity house light (during which time lever pressing led to the delivery of a reward pellet) was progressively decreased until it reached 3 s. During this phase of the experiment mice could make 4 types of responses, all of which led to the end of the current trial: dark (pressing during the dark phase), premature (pressing during the "dim" illumination phase), correct (pressing during the "full" intensity phase) and omission (not pressing the lever at all). Mice were trained until performance was stable at >50% correct responses per session. Mice were subsequently tested during a sham session where they were connected to a patch cable attached to a swivel at the top of the operant chamber or a stimulation session (the sham/stim order was counterbalanced across mice) where they were attached to the same patch cable and 473-nm blue light was delivered in 10-ms pulses at 20 Hz from a laser connected to the patch cable inside the operant box. The light pulses were delivered for the entire test session. Following the stimulation session mice were then trained on the high demand version of the task, the protocol of which was identical except for the house light being "bright" rather than "dim" during the anticipatory phase. Mice were again trained until performance was stable at >50% correct responses per session and were subsequently tested during laser stimulation in the same manner as described above. The data were collected using Med PC IV (v 4.2) and analyzed in GraphPad Prism (v 9.0).

## Von Frey test

Mechanical sensitivity was assessed using von Frey filaments of increasing force (0.04–1 g) applied manually to the plantar surface of one hind paw. Withdrawal frequencies were recorded with a 30-s interval between each of the five applications per filament. The investigator was blinded to the identity of the animals.

## Capsaicin injection

Acute secondary mechanical hypersensitivity was induced in the plantar area of the hind paw by subcutaneously injecting capsaicin in the lower hind leg. Illumination was conducted for 15 min at 15 min after capsaicin injection in opsin-expressing and control animals, as specified in each experimental scheme depicted in the figures. Mechanical sensitivity was measured before and 15 and 45 min after injection, in the absence or presence of illumination.

## Analysis of cFos expression following capsaicin injection and von Frey filament application

For four consecutive days mice were habituated to the experimenter and testing environment by being handled for 5 min and subsequently placed onto the von Frey testing apparatus for 30 min (days 1–3) and then 60 min (day 4). On day 5, in an alternating manner mice were randomly assigned to either receive a capsaicin injection (as described above, $N = 6$) followed 15 min later by von Frey testing (as described above) or to be exposed to the testing environment for the same amount of time but without capsaicin injection or von Frey testing ($N = 6$). Ninety minutes after the end of the von Frey testing/control exposure, mice were transcardially perfused and brains removed. Fifty μm sections were taken through the Cg1/2 and processed for cFos IHC. Confocal images were taken through the Cg1/2 at the co-ordinates form bregma AP +0.00, +0.70, +1.20, +1.50 mm. Quantification was performed on the maximum intensity projection of the z-stack. During perfusion, imaging and quantification the experimenter was blind to the experimental condition. A cell was considered cFos+ if it had a clear morphology and had an intensity that was > twice that of the background level[80].

## In vivo Ca²⁺ imaging combined with optogenetic stimulation and von Frey stimulation

Four weeks after GRIN lens implantation, mice were habituated to head fixation on a metal grid under a two-photon microscope (Leica TCS SP5 MP) for at least 1 week. After habituation, imaging data (256 × 256 pixels) were acquired using the microscope equipped with a 10X objective (LC PL APO 10x/0.40 CS2) at a frame rate of 20 Hz (920-nm wave length). The mice were awake and head-fixed during the data acquisition. The imaging experiment consisted of three 15 min-imaging sessions with a 15 min interval between sessions. In the second session, the mice received optogenetic stimulation. For the optogenetic stimulation, the mice implanted with the optic fiber were connected to optical patch cables (Thorlabs) coupled to a 633 nm laser (CrystaLaser LC). The light stimulation through the fiber was delivered at 20 Hz, 10 ms pulses (~4 mW/mm² intensity at the tip of optic fiber). The light stimulation was started with the first frame trigger provided by the microscope. In the first week of experiments, the mice received only optogenetic stimulation during imaging to record ongoing Ca²⁺ signals in the Cg1/2. In the following week, in addition to optogenetic stimulation, the mice received also mechanical stimulation. The mice exhibited less withdrawal behavior than mice without implantation. For the Ca²⁺ imaging experiments, the mice were head-fixed under the microscope restricting the movements of the mice compared to that of freely moving animals. In addition, the chronic implantation with a GRIN lens and an optic fiber is very likely to affect behavior. A previous study[81] showed that mice chronically implanted with cortical microelectrodes exhibited less withdrawal behavior than mice without implantation. Based on these, we used 0.6-g filaments in the Ca²⁺ imaging experiments. For the mechanical stimulation, a von Frey filament (0.6 g, 24 times, 1-s long with random intervals in each session) was applied to the plantar surface of the hind paw of the animal contralateral to the implantation site of GRIN lens during imaging in all sessions. The vF filament was attached to a servomotor. The servomotor was driven by a microcontroller (Arduino UNO) which is controlled by Bonsai software (v 2.4.0)[82]. The application timing was recorded by Bonsai. Paw withdrawal frequency was recorded during simultaneous Ca²⁺ imaging in each session. The investigator was blinded to the identity of the animals.

## Image processing and analysis for Ca²⁺ imaging

All analysis for Ca²⁺ imaging data was performed using scripts written in Python 3. Our Python code was written using NumPy[83] v1.18.1, SciPy[84] v1.4.1, statsmodels[85] v0.11.1, scikit-learn[86] v0.22.2.post1, pandas[87] v1.0.3, matplotlib[88] v3.2.1, and seaborn[89] v0.10.1.

Ca²⁺ imaging data were exported as tiff files using Fiji ImageJ and were motion corrected using NoRMCorre[90] algorithm implemented in CaImAn python package (1.8.4)[91]. Regions of interest (ROIs) were individually identified in each session using a constrained non-negative matrix factorization (CNMF) implemented as a function of CaImAn python package. To find the neurons detected in all sessions, the ROIs in each session were matched using register_multisession function of the CaImAn package. The ROIs that matched across all sessions were confirmed by visual inspection. Mismatched ROIs and ROIs that did

not correspond to neurons were excluded from the analysis. Fluorescence change ratio ($\Delta F/F0$) of ROI $i$ around vF stimulus was calculated as follows: $\Delta F/F0 = (f_i - f_{b,i})/f_{0,i}$. $f_i$ denotes fluorescence of ROI $i$ extracted through the CNMF process. $f_{b,i}$ denotes baseline fluorescence of the ROI $i$ and was calculated as a mean fluorescence for 0.5 s before onset of vF stimuli. $f_{0,i}$ was calculated as 8th percentile of the sum of the fluorescence of the ROI and the estimated background signal that was overlapped with the ROI i over a 1-s moving window.

The vF-evoked response of a neuron was calculated as the mean of $\Delta F/F0$s aligned to the onset of the vF stimuli. AUCs were obtained from the vF-evoked response in a time window from 0 to 5 s, from 0 to 1 s (during vF stimulation) and from 1 to 5 s (post vF stimulation) after the onset of the vF stimuli. AUC changes relative to session 1 were calculated as follows: $(\text{AUC}_{session2or3} - \text{AUC}_{session1}) / \text{absolute value of AUC}_{session1}$. To identify L2/3 and L5 neurons, positions of cells were calculated based on the actual ML coordinate of the GRIN lens identified in the histological analysis. To define cells that significantly responded to the vF stimuli, bootstrap analysis was performed for each neuron. In the analysis, the neural activity was circularly rotated (1000 random shifts) relative to time points when vF stimuli were applied, and the mean signals aligned to the time points of vF stimuli were calculated from the rotated data. If real vF-evoked response of a neuron was greater or less than the 97.5th percentile of the mean signals calculated from the 1000 randomly rotated data, the neuron was considered to be significantly responsive to the vF stimuli. For k-means clustering[86], the vF-evoked responses from 1 s before to 2 sec after the onset of the vF stimuli were binned (0.1 s bin) and the binned responses from three sessions were concatenated for each neuron. Principal components analysis was performed on the concatenated responses. The concatenated responses of all neurons from both control and ChrimsonR animals were projected on the first four principal components (>60% variance explained), and then clustered into two groups using k-means clustering with Euclidean distance measure. The optimal number of clusters was computed by using Silhouette index[92]. The groups were characterized by the following features: activated neurons across sessions (vF-activated neurons) and inhibited neurons (vF-inactivated neurons).

### Reporting summary

Further information on research design is available in the Nature Portfolio Reporting Summary linked to this article.

## Data availability

The data generated in this study are provided in the Source Data file. Raw imaging data are available on request from the corresponding author because of the large size of data. Source data are provided with this paper.

## Code availability

Our code is available in the following repository. For electrophysiology, https://github.com/Yu-Chao-Liu/HEKAexport or https://zenodo.org/records/6877792[93], https://github.com/Yu-Chao-Liu/IntrinsicProperties or https://zenodo.org/records/10700986[94], https://github.com/Yu-Chao-Liu/Intersections or https://zenodo.org/records/10702432[95]. For the other experiments, https://github.com/NUtaHei/Utashiro_NatCommun2024, or https://zenodo.org/records/10710076[96].

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

## Acknowledgements

We thank Drs. Linette Liqi Tan and Rohini Kuner for giving advice with the behavioral evaluation of the nociceptive response, Ulrike Amtmann for preparing solutions for the electrophysiological experiments, and Dr. Elke Fuchs for helping with the animal license (G-157/16, G-131/19). We thank and acknowledge the staff and services offered by the Light Microscopy Facility at the DKFZ. The project was funded in part by SFB1158 grant (project B06) from the Deutsche Forschungsgemeinschaft (DFG) to H.M.

## Author contributions
The project was coordinated by H.M. The virus-tracing experiments were executed by K.Y., N.U. and B.W. The in vitro electrophysiological experiments were performed by Y.-C.L. The behavioral studies were performed by K.Y. and D.A.A.M. The calcium imaging experiments were performed by N.U. The manuscript was written by N.U., D.A.A.M. and H.M. with the contribution of the other authors.

## Funding

## Competing interests
The authors declare no competing interests.
