## [Peer Review File · Nature Communications]

Long-range inhibition from prelimbic to cingulate areas of the medial prefrontal cortex enhances network activity and response executionREVIEWER COMMENTS

Reviewer #1 (Remarks to the Author):

This study by Utashiro and colleagues discovers and investigates an inhibitory long-range projection from prelimbic PFC (PrL) to anterior cingulate cortex (ACC). Using viral anterograde and retrograde tracing, they describe a GABAergic projection that appears to originate in layer 1 (L1) of PrL and projects to ACC L1. In line with the laminar distribution of these axons, they reveal L1 interneurons as the main targets of this projection using optogenetics in brain slices. Moreover, optogenetic activation of the projection is shown to enhance L5 pyramidal neuron activation by disinhibition. Next, the functional role of the inhibitory projection is addressed for 2 separate behavioral tasks. In a simple visual attention paradigm, optogenetic activation decreases correct while increasing premature responses, suggesting that the animals displayed greater impulsivity. In a mechanical stimulus detection paradigm, optogenetic activation increases responsivity across a wide range of stimulation intensities. Using an innovative combination of optogenetic activation and GRIN lens based deep calcium imaging in behaving mice, the authors go on to show that activation of the PrL afferents causes a net increase of the responses to mechanical stimuli, that appears to be localized to L5 pyramidal cells. Conversely, ongoing spontaneous activity was not affected. In line with the interpretation that this effect is mediated by disinhibition via L1 interneurons, optogenetic stimulation selectively enhanced excitatory but not inhibitory responses to mechanical stimuli.

This study adds an important facet to our rapidly growing knowledge of long-range inhibitory circuits that was to a large extent produced by the senior author's work over the last years. This is crucial since most functional interpretations and circuit models of brain function still ignore these 'non-canonical' pathways. Moreover, as this study beautifully illustrates, long-range inhibition has unique organization and function. The manuscript is written in balanced fashion, acknowledging and integrating work from recent to classic studies in the field, the experiments are generally done to high standards (but see below), and the data are potentially of great importance to the neuroscience community.

However, there are a number of experiments, analyses and clarifications on experimental approaches and data analyses that need to be fully addressed before the validity of the data and conclusions can be judged.

Major:

The data in Fig. 1 and ED Fig. 1 convincingly show that the GABAergic projection exists, but there are a number of important open questions that are at present not addressed here. First, what is the laminar distribution of the axons connecting PrL to ACC and vice versa? This needs to be addressed by additional experiments and quantifications. The images appear to indicate that the projection may preferentially

target superficial L1 (L1a). If so, that is intriguing, in particular since corticocortical excitatory projections are typically localized in L1b (e.g. Cruikshank et al., 2012, Journal of Neuroscience). This may therefore add an important organizational difference between excitatory and inhibitory long-range communication that – if present – should be discussed.

Along similar lines, the laminar localization of the GABAergic neurons that project from PrL to ACC seems to be exclusively in L1 (based on the example images). This is an interesting finding which deserves to be addressed both qualitatively and quantitatively in the paper. They should quantify the position of retrogradely-labeled cell bodies across cortical depth for ACC-projecting GABAergic PrL neurons (to add to Fig. 1d-f) and for PrL-projecting GABAergic ACC neurons (to add to ED Fig. 1d-f).

The experiments in Fig. 2g to k need to be explained in far greater detail. They may well make sense, but it remains completely unclear in Fig. 2h,i what was recorded and which traces are the result of subtraction. What does 'EPSCs and concomitant IPSCs' mean here? Are those the recorded EPSC-IPSC sequences (if so, how do you tell them apart)? And how do you then manage to record a pure EPSC at -50 mV? And how do you scale the EPSC between -50 and -92 mV? Please take the time to fully explain the experiment in the main text or the figure legend.

As far as I can see, the distinction between pSBCs and pNGFs is based purely on physiology. While that is fine for the present work, then please remove one or both of the morphological reconstructions from Fig. 2. The problem at present is that the L1 interneuron categorization appears to be based also on morphology, yet no quantification is provided. Indeed, the nomenclature of L1 interneuron types has arguably not been settled, and while this is not an issue for their work, they should take care not to convey an incomplete picture on this.

The analyses in Fig. 6 are important but incomplete. For instance, it is currently not clear how calcium responses evolve across trials (with and without optogenetics) and how the responses of a given neuron are affected by optogenetics. Please include plots to address both of these points for Fig. 5 and Fig. 6, such as scatterplots comparing the responses in individual neurons across sessions (to address point 2). Moreover, please plot how the responses between control and Chrimson animals compare for vF-activated L2/3 neurons and L5 neurons. The same question applies to the vF-inactivated L2/3 and L5 populations.

Minor:

The introduction needs some work. For instance, the arguments in lines 31-32 and 35-36 repeat. Moreover, please clearly motivate the focus on the PrL to ACC projection, in particular since several

additional GABAergic projections are described in the study. The sentence lines 45-48 is hard to understand, I suggest to re-write.

Please motivate the choice of analysis time windows in Fig. 5 and 6. Was mechanical stimulation applied for 1s, or is there another reason for this choice?

Line 99: typo GABAergic

Lines 331-336: One obvious candidate system for recruiting L1 interneurons is acetylcholine, in case the authors wish to mention that.

Fig 5e-k: While it is commendable that they show the data from all 3 sessions for both control and ChrimsonR animals, this data display somewhat dilutes the core message here. I leave it up to the authors, but they could move part of this to the extended data, and focus more on the effect of optogenetic stimulation here.

Please add an example trace to Fig. 2 to show the effect of GABAAR blocker application on light-evoked IPSCs for patched pSBC neurons.

ED Fig. 3 shows that the effect persists in the post-stimulation period (unlike in session 3 for Fig. 4). Please provide a rationale for how you interpret this result in cases with or without capsaicin.

The labeling in ED Fig. 7e-f is confusing, because at first glance it looks like 7e is the 'no withdrawal' graph and 7f is the 'withdrawal' graph. Please adjust the legend positioning to make it clearer that 7e is the 'control' graph and 7f is the 'Chrimson' graph.

Reviewer #2 (Remarks to the Author):

The manuscript by Utashiro et. al. reports cortical projecting GABAergic inhibitory cells in cortical communication within frontal cortical areas. The authors 1) described the presence of GABAergic

neurons in prelimbic cortex (PrL) projecting anterior cingulate cortex (ACC), 2) mapped the connections to different types of neurons in ACC and 3) reported the disinhibition of L5 pyramidal cells in AAC via the long-range GABAergic projection neurons from PrL to AAC enhance motor execution to external stimuli.

The current study mapped another GABAergic projecting circuit motif in the frontal cortical area and reported the significance of GABAergic projection from PrL to AAC in animal's behavior. For that, the authors used a range of methods including in vitro electrophysiology to delineate the disinhibitory connection and in vivo calcium imaging to characterize the role of this circuit motif. While the current study provides an interesting aspect of cortical projecting GABAergic cells in the frontal cortical area, it seems some of the key control experiments are missing and some of the results, especially L5 pyramidal cell activity during vF stimulation, are somewhat confusing and unclear.

1. To disinhibit L5 pyramidal cells in AAC by PrL GABAergic cells, one of the key assumptions is that both the projecting PrL GABAergic cells and their target GABAergic cells in AAC are active during vF stimulation and/or a visual attention task. Please provide evidence these cells are active during vF stimulation or visual attention task.

2. The authors identified the types of GABAergic cells in AAC that PrL GABAergic cells target are SBCs and NGFs. Yet, it is not clear what type of GABAergic cells in PrL are long-range projecting GABAergic cells to ACC. Identifying the projecting GABAergic cell types will be important to understand the complete circuit diagram.

3. The authors used a viral approach to identify the somatic location of PrL GABAergic cells and their target areas. The presented data related to the presynaptic cells and their axon innervation to other areas in the manuscript are a couple of example images but there is no quantification. Please provide a quantitative analysis of somatic location of projecting GABAergic cells in PrL and their axonal distribution in ACC and other areas. The authors commented these axons are detected in Extended Figure 2 but it is not clear how robust this is. Given PrL and AAC are elongated structures within the frontal cortical area spanning a broad range of AP axis, it will be informative to know the location of presynaptic cells and the areas where these axons innervate.

4. The authors provided thoughtful comments in the Discussion on the potentially opposite net effect on ACC L5 pyramidal cells by PrL GABAergic cell-mediated inhibition of SBC and NGF in ACC. As the authors pointed out, at least in connection probability and IPSCs, SBC are more strongly connected by PrL GABAergic cells which may lead to inhibition of L5 pyramidal cells. However, current Ca⁺⁺ imaging data show net excitation of L5 pyramidal cells upon vF and optogenetic stimulation. One possibility might be that the spontaneous activity of SBC and NGF are different – NGF may have higher spontaneous activity and be more strongly driven by vF stimulation. Please provide spontaneous activity of SBC and NGF cells

in ACC. How the SBC and NGF in ACC are differentially driven by vF-stimulation will be informative to understand how the disinhibition between two different cortical areas by projecting GABAergic cells.

5. Figure 2c: My understanding of the 5HT3R positive GABAergic population in cortex is that SBC and NGF cells are subgroups within the 5HT3R population. Please clarify whether NGF and SBC are 5HT3R negative populations or 5HT3R in the current study indicate non-NGF and non-SBC 5HT3R-positive populations.

6. Figures 5&6: the authors conducted simultaneous single cell Ca⁺⁺ imaging with optogenetic stimulation. I could not find any description on how the authors controlled the crosstalk between optogenetic stimulation light and scanning light. Please provide clear control data indicating there is no crosstalk.

7. Figure 5: It is not clear why light stimulation reduced the vF stimulation-evoked responses in the control group. Given that AAV-FLEX-tdTomato was expressed in the control group, the optogenetic stimulation should not evoke any changes. Please provide an explanation on these results.

8. Figure 5: The vF stimulation-induced activity of L5 pyramidal cells were still significantly enhanced in session 3 even after an optogenetic stimulation session without light stimulation. Related to this issue, even at a single cell level (Figure 6), vF-activated cells show elevated response even after optogenetic stimulation. Please provide an explanation why the vF stimulation-induced activity is still high in session 3 without optogenetic stimulation. Is it some sort of synaptic plasticity?

9. Figure 6: Figure 5 results indicate optogenetic stimulation of PrL GABAergic terminals in ACC only increases L5 pyramidal cell activity during vF stimulation. It is not described whether the single cell analysis in Figure 6 are cells from all layers or from L5. If the data were pulled from all layers, please provide a relationship between ensembles (1 & 2) and layers (L2/3 and 5).

Minor comments)

10. In vitro electrophysiology lacks the number of animals that are used in the study. Please report the used animal numbers

11. Extended Figure 7b: values for scale bar are missing.

12. line 319: Jiang et al., 2015 (PMID: 26612957) can be added here.

Reviewer #3 (Remarks to the Author):

In this study by Utashiro et al., the authors identified GABAergic projections from the prelimbic (PrL) cortex to the anterior cingulate cortex (ACC), primarily originating from layer (L) 1. The authors investigated which ACC neurons are targeted by PrL and found that the projections are exclusively to L1, particularly to neurogliaform (NGF) cells and single-bouquet cells (SBCs). Furthermore, the authors have evaluated the functional impact of PrL GABAergic projection on ACC circuit revealing that these projections likely regulate L5 pyramidal neurons (PNs) activity via L1 interneurons (INs). The study reveals the function of long-range projection cortico-cortical GABAergic INs, both anatomically and functionally. The results found in the study reveal a different mechanism of top-down control that could be working in parallel (or even competing) with excitatory glutamatergic projections. Additionally, showing a relationship between the investigated projections and different behavioral paradigms is quite interesting and an important contribution to the field of systems neuroscience. However, there are a number of issues that need to be addressed:

1. In Fig. 1

- a. It would be nice to quantify the ratio of inhibitory projections to excitatory, i.e. how many of the projections are actually inhibitory in L1. Could be done using a Syn or CamKII promoter based injection in PrL using a different fluorophore and quantifying excitatory vs inhibitory targeting in L1 of ACC.
- b. Please clarify in the schematics that the animals are GAD-cre
- c. Quantify projection density across layers to show L1 specificity in ACC.
- d. Fig 1 b, e- it is hard to tell the specificity of the injection to the regions, show an image of the whole section to better appreciate the injection sites in PrL and ACC.

2. In Fig. 2

- a. Raw data examples for the reconstructed morphologies could be nice in supplementary. An important concern is the distinction between NGFs and SBCs. While the authors state that they identify NGFs by first spike latency, they show no quantification for that! FSL should be quantified and be compared between NGFs and SBCs, especially since the data is there! Include criteria for NGF vs SBC classification-intrinsic properties table to be included
- b. In the discussion the authors claim that NGFs are targeted less than SBCs (line 329), such a claim should be backed by stats in the results, which is not the case. In fact, Fig. 2c shows that its unlikely there's a significant different.

c. The authors mention in the methods how they identified SOM and PV INs electrophysiologically but this data is never shown, please include it in supplementary.

d. Why are they showing the GABAaR blocker SR95531 only for the NGFs, was it only tested on NFGs?

3. In Fig. 3

a. Include in the schematic the YFP injections in addition

b. The authors never mentioned the duration of the stimulus, nor in which epoch they stimulate. Not in the figure, not in the main text, and not in the method! Also, what is the percentage of stimulated trials per session? Please clarify in the text/methods and the schematic.

c. In c-e the legend for red is ChR2, however there's no ChR2 in sham! Please clarify what are the sham conditions-is it no light stimulus or a light stimulus of a different frequency?

d. In the figure caption and methods, the authors mention that there are 3 types of responses, however in the main text they mention 4 types of responses. The authors should specify that trials with responses in the dark are omitted.

e. The authors use GAD-Cre mice, hence the GABAergic projections that are stimulated could be coming from anywhere. Understandably, in Fig 1, it was demonstrated that these cells primarily reside in L1, forming a novel L1-L1 connectivity. However, to really show that this circuit is involved in this behavior, the authors should demonstrate this by using a L1 specific Cre line, such as NDNF.

4. Fig. 4:

a. Would be nice to include a short sentence explaining von Frey filaments in the main text, not everyone might be familiar with it.

b. The schematic for the von frey filament seems like an afterthought, it is not very informative. Please add more information to understand the experiment

c. Show a comparison of the von frey response across sessions in both control and optogenetic conditions

5. Fig. 5: in this figure and figure 6 and related supps axis have very small font! You can barely read it!

a. Demonstrate that the grin lens implantation does not cause disruption of L1 fibers. Do a quantification and include images of the brains after perfusion to show the intactness of L1 in these experiments.

b. Why was 0.6gms used in these experiments? According to fig 4, 0.6gms had no difference with and without optogenetic stimulation. Please clarify.

c. The schematic is not very informative with respect to the results- when was the optogenetic stimulation delivered? Elaborate on the timing within the schematic. This goes for all the figures, it is not very clear when/how long the stimuli lasted.

- d. Please put comparisons and level of decrease for each in the main text (numerical values and statistics).
- e. For L2/3 and L5 responses, show example traces. From the confocal image example, there doesn't seem to be any expression of GCaMP6 in L2/3. How many cells were imaged in L2/3 vs L5. Could the no difference be merely due to low number of imaged cells?
- f. What about responses differences in L1 cells themselves? Does vF cause an inhibition in these cells? In order to decipher circuit mechanism of this phenomenon the authors should also record responses of L1 Ins themselves.

Minor points

It appears the manuscript was written by multiple people- which makes the writing style different in different parts of the manuscript. Please unify the writing styles as well as ensure the manuscript has been proofread for grammatical errors/usage

- a. The authors should address every subpanel for every figure in the main text. Figure 3 has 5 panels but only referenced in the main text as Fig. 3
- b. In Fig 6, use a different color (not red) to highlight vF activated neurons. At the moment it is confusing to use red since previously it denoted the chrimson expressing ones
- c. In extended data Fig. 6. The authors should compare control vs chrimson rather than chrimson in different groups! Also they should compare optogenetic activity and baseline activity in small windows of time at a single neuron level rather than a population and averaging over a whole sessions! Using these analysis, if there was a crosstalk it will obviously average out!
- d. Calling blocks before, during, and post optogenetic "sessions" is very confusing, as in between sessions implies to me that the animals were returned to animal facility. But I guess this is personal point of view.
- e. A reference should be cited in line 314 (... their recruitment shapes the activity of local circuits).
- f. In line 318 authors should state what they mean by distinctly wired, and why this wiring is important to their findings.
- g. In line 320 , the use of the word "master-regulators" is too strong. They are gatekeepers, and control the flow of excitation, but master regulators implies something that is very broad and controls every aspect of information processing.
- h. In line 321 the authors claim that these projections control neuronal signal across layers, but also in the result they claim (and show) that the control is specific to L5 and not L2/3. Please clarify?
- i. The authors mention in the results injection in medial orbital cortex (MO; supp. 2) but in the methods no injection site is mentioned for MO. If it was off target injections of

PrL there's no harm in mentioning that, perhaps mentioning the depths of these off-target injections could save animals in other labs!

j. Overall, with so many injections with different viral vectors x transgenic animals and different injection volumes, and different brain areas, would be nice to have supplementary summary table.

Reviewer #4 (Remarks to the Author):

The authors identified GABAergic projections connecting two parts of the mouse medial prefrontal cortex (mPFC), namely a part they call prelimbic cortex (PrL) and a part they call anterior cingulate cortex (ACC). They also identified the neurons targeted by these projections as being single-bouquet cells and neurogliaform cells. Furthermore, they performed a functional characterization of

The authors use anterograde and retrograde tracing and identified reciprocal GABAergic projections between PrL and ACC. They found layer I to be the mainly targeted layer. They performed patch clamp recordings to identify the cells in ACC being targeted by the PrL GABAergic efferents, and vice versa. In both areas, the GABAergic projections selectively inhibited putative single-bouquet and neurogliaform cells. Furthermore, the authors determined that stimulation of PrL GABAergic projections in the ACC resulted in an increased impulsivity and affected the performance of mice in an operant test of sustained attention. Additionally, the authors determined that optogenetic activation of these projections increased sensitivity and response of mice to mechanical stimulation. The authors also determined that recruitment of PrL GABAergic terminals in the ACC enhanced the activity of layer V neurons.

These excellent results are unfortunately accompanied by a deplorably low degree of anatomical accuracy which must be corrected not only for this study to really advance our understanding of medial prefrontal circuitry, but also to enable reproducibility of the results. The authors state that the human and mouse mPFC comprise several subregions, including PrL and ACC. The term prelimbic cortex is used in rodent, but not primate research. However, the authors use it indistinctively for both species. Specifically, PrL designates an architectonically distinct area (not a subregion or region) in the rodent brain considered to be homologous to area 32 in the brains of humans and non-human primates (Preuss et al., 1995). The term ACC is used in functional imaging studies to designate a region of the human cingulate cortex encompassing Brodmann's areas 24, 25, 32 and 33. Given the drawings provided in Fig. 1a and 1d, the area designated as ACC by the authors actually corresponds to the portion of cortex occupied by the dorsal anterior cingulate area (ACd, or area 24d) and the ventral anterior cingulate area (ACv, or area 24v) of Preuss et al. (1995). And the injection site could actually be at the border between these two areas. Areas ACd and ACv correspond to areas cg1 and cg2 of Franklin and Paxinos (2013), respectively. Thus, the term ACC is not synonym of ACd and ACv. Rather, ACC encompasses areas IL, PrL, cg1 and cg2. Furthermore, areas cg1 and cg2 have each been subdivided into a rostral (cg1/cg2) and a caudal (cg1'/cg2') area (Vogt& Paxinos, 2014; van Heukelum et al., 2020). In which of these four areas

(cg1, cg2, cg1', cg2') did the authors inject the retrograde tracer? This must be specified in order to enable a correct interpretation of the results. Ideally, the authors should provide a series of overview images of sections containing the injection site. These could be either a low-resolution fluorescent image of the entire section, or a bright-field or dark-field image of the sections in question (as was done, e.g., in Fig. 7 of Király et al., 2020). These images should be accompanied by high resolution micrographs of sections enabling architectonic identification of the site.

Is the area they designate PrL really area PrL? Could it be part of area M2? In figure 1d area PrL clearly extends onto the dorsal surface of the hemisphere, although this area is generally restricted to the mesial surface of the hemisphere. Also here, the authors should provide overview images of sections containing the injection site and high resolution micrographs of sections enabling architectonic identification of the site.

Minor points:

I find it surprising and an example of poor citation practice that the authors have not cited a single article by Brent Vogt, one of the neuroanatomists who has most contributed to our understanding of the architectonic segregation of the cingulate cortex across species.

A table with information concerning the exact number of animals included in each experimental group would be helpful.

Refs.

Franklin & Paxinos (2013) *The mouse Brain in Stereotaxic Coordinates*. Amsterdam, Academic Press

Király et al. (2020) *Nat Commun* 11: 4686. <https://doi.org/10.1038/s41467-020-18472-y>

van Heukelum et al., (2020) *TINS* 43: 285-299

Vogt & Paxinos (2014) *Brain Struct Funct* 219: 185-192

We thank the Reviewers for the critical and constructive comments that we address below point by point. The revision process was fairly long for three reasons: Firstly, we had to generate mice of different genetic background to perform some of the requested experiments. Unfortunately, during the Coronavirus pandemic the size of the mouse colonies was reduced so drastically that even now we are not back to the status before the pandemic. Secondly, we performed additional *in vitro* electrophysiological experiments and thirdly, we performed behavioral experiments combined with cFos staining that were not requested, but that answered a question that we could not address at least in part. The main messages of the manuscript have not changed, but the Reviewers have helped us to generate a revised version in which mistakes have been corrected, ambiguities have been clarified and additional data corroborate some of the previous findings. We hope that the current version finds the Reviewers' approval.

Reviewer #1 (Remarks to the Author):

This study by Utashiro and colleagues discovers and investigates an inhibitory long-range projection from prelimbic PFC (PrL) to anterior cingulate cortex (ACC). Using viral anterograde and retrograde tracing, they describe a GABAergic projection that appears to originate in layer 1 (L1) of PrL and projects to ACC L1. In line with the laminar distribution of these axons, they reveal L1 interneurons as the main targets of this projection using optogenetics in brain slices. Moreover, optogenetic activation of the projection is shown to enhance L5 pyramidal neuron activation by disinhibition. Next, the functional role of the inhibitory projection is addressed for 2 separate behavioral tasks. In a simple visual attention paradigm, optogenetic activation decreases correct while increasing premature responses, suggesting that the animals displayed greater impulsivity. In a mechanical stimulus detection paradigm, optogenetic activation increases responsivity across a wide range of stimulation intensities. Using an innovative combination of optogenetic activation and GRIN lens based deep calcium imaging in behaving mice, the authors go on to show that activation of the PrL afferents causes a net increase of the responses to mechanical stimuli, that appears to be localized to L5 pyramidal cells. Conversely, ongoing spontaneous activity was not affected. In line with the interpretation that this effect is mediated by disinhibition via L1 interneurons, optogenetic stimulation selectively enhanced excitatory but not inhibitory responses to mechanical stimuli.

This study adds an important facet to our rapidly growing knowledge of long-range inhibitory circuits that was to a large extent produced by the senior author's work over the last years. This is crucial since most functional interpretations and circuit models of brain function still ignore these 'non-canonical' pathways. Moreover, as this study beautifully illustrates, long-range inhibition has unique organization and function. The manuscript is written in balanced fashion, acknowledging and integrating work from recent to classic studies in the field, the experiments are generally done to high standards (but see below), and the data are potentially of great importance to the neuroscience community.

However, there are a number of experiments, analyses and clarifications on experimental approaches and data analyses that need to be fully addressed before the validity of the data and conclusions can be judged.

Major:

The data in Fig. 1 and ED Fig. 1 convincingly show that the GABAergic projection exists, but there are a number of important open questions that are at present not addressed here. First, what is the laminar distribution of the axons connecting PrL to ACC and vice versa? This needs to be addressed by additional experiments and quantifications. The images appear to indicate that the projection may preferentially target superficial L1 (L1a). If so, that is intriguing, in particular since corticocortical excitatory projections are typically localized in L1b (e.g. Cruikshank et al., 2012, J. Neurosci.). This may therefore add an important organizational difference between excitatory and inhibitory long-range communication that – if present – should be discussed.

Following the Reviewer's suggestion, we performed additional experiments and quantifications of the laminar distribution of GABAergic axons between the PrL and the ACC (renamed Cg1/2 according to the suggestions of Reviewer 4). The results are now shown in Fig.1 and Supplementary Fig.3.

Indeed, PrL and Cg1/2 GABAergic projections were denser in L1a compared to L1b. The previous study referred to by the reviewer (Cruikshank et al., 2012, J. Neurosci.) showed that matrix thalamocortical projections were concentrated in L1a, while corticocortical excitatory projections were denser in L1b in the medial PFC. In our results, however, the density of excitatory axons was comparable between L1a and L1b. There are multiple differences regarding the excitatory projections in the Cruikshank et al. study and ours that may account for this, including age (previous study: postnatal days 13 - 17, current study: older than 7 weeks), different types of viruses or different coordinates of the injection sites.

Nevertheless, the organizational difference in excitatory and inhibitory projections in L1 are evident, but the functional significance thereof is not clear. At the moment it is not possible to infer from the anatomical organization how the function of our neurons relates to that of glutamatergic input onto L1 INs. We now mention this in the discussion and cite the Cruikshank et al. study.

Along similar lines, the laminar localization of the GABAergic neurons that project from PrL to ACC seems to be exclusively in L1 (based on the example images). This is an interesting finding which deserves to be addressed both qualitatively and quantitatively in the paper. They should quantify the position of retrogradely-labeled cell bodies across cortical depth for ACC-projecting GABAergic PrL neurons (to add to Fig. 1d-f) and for PrL-projecting GABAergic ACC neurons (to add to ED Fig. 1d-f).

In the original manuscript, we showed some examples of retrogradely labelled cells. This led to the confusion that the cell bodies of PrL GABAergic projection neurons were exclusively located in L1. However, as shown for Cg1/2 GABAergic projection neurons in Extended Data Fig.1f in the original manuscript, we detected cell bodies of PrL GABAergic projection neurons not only in L1 but also in other layers. As the retrograde AAV that we used in the original study was suboptimal (few cells were labeled) and hence precluded a quantitative analysis, we instead injected CTB into the Cg1/2 or the PrL of GAD^{EGFP} mice and performed additional retrograde tracing. Consistent with the results by retrograde AAV, the cell bodies of GABAergic projection neurons were located in all cortical layers as is now shown in Supplementary Fig. 1 and 4.

The experiments in Fig. 2g to k need to be explained in far greater detail. They may well make sense, but it remains completely unclear in Fig. 2h,i what was recorded and which traces are the result of subtraction. What does 'EPSCs and concomitant IPSCs' mean here? Are those the recorded EPSC-IPSC sequences (if so, how do you tell them apart)? And how

do you then manage to record a pure EPSC at -50 mV? And how do you scale the EPSC between -50 and -92 mV? Please take the time to fully explain the experiment in the main text or the figure legend.

We followed the Reviewer's advice and provide detailed descriptions in the Methods section. What was previously termed 'EPSCs and concomitant IPSCs' is now referred to as 'compound PSCs', which include EPSC-IPSC sequences and putative EPSCs. Compound PSCs and putative pure EPSCs were recorded at ~-50 mV and -92 mV, respectively. Feedforward IPSCs are the result of subtraction. The initial slope of compound PSC was scaled to the initial slope of pure EPSC recorded at -92 mV and subtracted by the pure EPSC recorded at -92 mV. To minimize the IPSCs derived from the antidromically activated GABAergic neurons, the stimulating electrode was placed at least 800 μm away from the recorded cell and the intensity was tuned to a level which did not always elicit EPSC-IPSC sequences recorded at near -50 mV, commensurate with the spike probability of INs that was always below 100% (Fig. 2d,e). Therefore, some putative pure EPSCs were observed at near -50 mV. In addition, the data were discarded if the latency of onset of the feedforward IPSC and of the EPSC was shorter than 1 ms. In our dataset, the average latency was 3.03 ± 0.45 ms ($n = 12$ cells). It is unlikely that antidromically evoked IPSCs constituted a major component of isolated feedforward IPSCs.

As far as I can see, the distinction between pSBCs and pNGFs is based purely on physiology. While that is fine for the present work, then please remove one or both of the morphological reconstructions from Fig. 2. The problem at present is that the L1 interneuron categorization appears to be based also on morphology, yet no quantification is provided. Indeed, the nomenclature of L1 interneuron types has arguably not been settled, and while this is not an issue for their work, they should take care not to convey an incomplete picture on this.

We understand the Reviewer's point of view, but find it a pity if we do not provide any information regarding the morphology. Thus, we have added 11 more reconstructions from 5 pSBCs and 6 pNGFs and provide the normalized axonal density plots in Fig. 2a. We provide a figure for this Reviewer (Fig. 1 for Reviewer 1), and ask whether he/she wishes that we include it as yet another Supplementary figure. Needless to say, if the Reviewer still wishes that we remove this panel, we will do so.

The analyses in Fig. 6 are important but incomplete. For instance, it is currently not clear how calcium responses evolve across trials (with and without optogenetics) and how the responses of a given neuron are affected by optogenetics. Please include plots to address both of these points for Fig. 5 and Fig. 6, such as scatterplots comparing the responses in individual neurons across sessions (to address point 2). Moreover, please plot how the responses between control and Chrimson animals compare for vF-activated L2/3 neurons and L5 neurons. The same question applies to the vF-inactivated L2/3 and L5 populations.

Following the Reviewer's suggestion, we first generated interaction plots of all neurons (see Figure 2 for Reviewer1). However, since there were too many data points to display in this way, we thought it is more informative to provide heat maps of the responses of individual neurons across sessions (Supplementary Fig. 13a and b). We have also provided responses from L2/3 and L5 vF-activated or -inactivated neurons in Supplementary Fig. 14 and Fig. 6.

Minor:

The introduction needs some work. For instance, the arguments in lines 31-32 and 35-36 repeat. Moreover, please clearly motivate the focus on the PrL to ACC projection, in

particular since several additional GABAergic projections are described in the study. The sentence lines 45-48 is hard to understand, I suggest to re-write.

We followed the Reviewer's suggestion and rephrased the sentences mentioned above. We think it is clearer now what motivated us to study this particular projection in more detail, that is at the functional level. We removed in the introduction the information that the connectivity is "bidirectional". This information is now in the result section. There we state that in other brain regions GABAergic projection neurons often connect brain areas bidirectionally, and we his in the current study as well. Furthermore, we have added in the revised version electrophysiological in vitro data showing that also in the opposite direction GABAergic projection neurons target L1 INs. Therefore, we think that this is a leitmotiv regarding the wiring principle of cortical GABAergic projection neuron. In other words, this connectivity pattern is not restricted to GABAergic projection neurons from the PrL to the Cg1/2, but can be found for other cortico-cortical GABAergic projection neurons.

In addition (but this explanation is meant only for the Reviewer as it would be somewhat awkward to mention this in the introduction), we decided to study the projection from the PrL to the Cg1/2 in more detail as the target area was more "suitable" to address the question whether the novel projections modulate behavior. In other words, several behavioral tests came to mind that could be employed.

Please motivate the choice of analysis time windows in Fig. 5 and 6. Was mechanical stimulation applied for 1s, or is there another reason for this choice?

Yes, the stimulation duration was 1s. To exemplify the details of the stimulation better, we updated the schematic of the procedure used for the Ca^{2+} imaging experiments, and hope that this is now clearer (Fig. 5e).

Line 99: typo GABAergic

Thank you for pointing this out. The typo has been corrected.

Lines 331-336: One obvious candidate system for recruiting L1 interneurons is acetylcholine, in case the authors wish to mention that.

We absolutely agree that for completion's sake the direct cholinergic input onto L1 INs should be mentioned, and we have done so in the discussion.

Fig 5e-k: While it is commendable that they show the data from all 3 sessions for both control and ChrimsonR animals, this data display somewhat dilutes the core message here. I leave it up to the authors, but they could move part of this to the extended data, and focus more on the effect of optogenetic stimulation here.

We very much prefer to leave these panels in the main manuscript, because they convey a message that we think is important, namely they readily show the direct comparison between the three sessions in the control group and in the ChrimsonR group. For instance, the reduction in the AUC between session 1 and 2 in the control group most likely reflects habituation, possibly pointing to an even larger increase than the one shown between session 1 and 2 in the ChrimsonR group. However, if the Reviewer insists, we are of course prepared to move the two panels to the supplementary figures.

Please add an example trace to Fig. 2 to show the effect of GABA_AR blocker application on light-evoked IPSCs for patched pSBC neurons.

We have added an example trace of light-evoked IPSCs recorded in a pSBC in the absence and presence of the GABA_AR blocker SR (Fig. 2a).

ED Fig. 3 shows that the effect persists in the post-stimulation period (unlike in session 3 for Fig. 4). Please provide a rationale for how you interpret this result in cases with or without capsaicin.

Capsaicin injections induce acute pain. Under acute pain, the vF stimuli are very likely to be perceived more strongly. This may lead to the development of long-lasting synaptic plasticity. Thus, indeed there was a left-shift of the curve in session 3 following capsaicin injection. This interpretation has been included also in the discussion.

The labeling in ED Fig. 7e-f is confusing, because at first glance it looks like 7e is the 'no withdrawal' graph and 7f is the 'withdrawal' graph. Please adjust the legend positioning to make it clearer that 7e is the 'control' graph and 7f is the 'Chrimson' graph.

Following the Reviewer's suggestion, we revised the figure (Supplementary Fig. 12e and f) and adjusted the positioning of the legend.

Reviewer #2 (Remarks to the Author):

The manuscript by Utashiro et. al. reports cortical projecting GABAergic inhibitory cells in cortical communication within frontal cortical areas. The authors 1) described the presence of GABAergic neurons in prelimbic cortex (PrL) projecting anterior cingulate cortex (ACC), 2) mapped the connections to different types of neurons in ACC and 3) reported the disinhibition of L5 pyramidal cells in AAC via the long-range GABAergic projection neurons from PrL to AAC enhance motor execution to external stimuli.

The current study mapped another GABAergic projecting circuit motif in the frontal cortical area and reported the significance of GABAergic projection from PrL to AAC in animal's behavior. For that, the authors used a range of methods including in vitro electrophysiology to delineate the disinhibitory connection and in vivo calcium imaging to characterize the role of this circuit motif. While the current study provides an interesting aspect of cortical projecting GABAergic cells in the frontal cortical area, it seems some of the key control experiments are missing and some of the results, especially L5 pyramidal cell activity during vF stimulation, are somewhat confusing and unclear.

1. To disinhibit L5 pyramidal cells in AAC by PrL GABAergic cells, one of the key assumptions is that both the projecting PrL GABAergic cells and their target GABAergic cells in AAC are active during vF stimulation and/or a visual attention task. Please provide evidence these cells are active during vF stimulation or visual attention task.

At the outset of this study, the question was, how this novel connectivity mediated by GABAergic projection neurons would affect the output cells in the target area and behavior involving/supported by the ACC (now termed Cg1/2 following the suggestion of Reviewer 4). To directly investigate the activity of L1 neurons during vF stimulation would constitute an entirely new study. A major hindrance is the sinus above the Cg1/2, and even imaging

employing GRIN lense plus a prism would be a great challenge. In the attached scheme for the Reviewer (Fig. 3 for Reviewer 2 and Reviewer 3), we would like to highlight the anatomical features that make imaging of L1 INs almost impossible in the Cg1/2 whereas in motor or visual cortex this would not be a problem.

The imaging of the source cells is a different challenge due to the scarcity of labeled cells when using retrograde viruses. We tried several kinds (retrograde AAVs and CTBs), and they indeed allowed nice anatomical tracing studies (as shown in Supplementary Fig. 1), but none were satisfactory for functional studies (i.e. labeling sufficient cells that enable calcium imaging).

However, we sought to address the Reviewer's question regarding the recruitment of L1 INs in a behavioral setting by performing a quantificational study of cFos expression in control and capsaicin treated animals. As this line of experiments was quite time consuming, we settled on a behavioral paradigm in which more robust activation of neurons is expected. Indeed, the data presented in Supplementary Fig. 7 show that L1 INs are activated more upon mechanical stimulation in capsaicin treated mice compared to control mice.

2. The authors identified the types of GABAergic cells in AAC that PrL GABAergic cells target are SBCs and NGFs. Yet, it is not clear what type of GABAergic cells in PrL are long-range projecting GABAergic cells to ACC. Identifying the projecting GABAergic cell types will be important to understand the complete circuit diagram.

To determine the molecular identity of GABAergic projection neurons from the PrL to the Cg1/2, we performed tracing experiments (Supplementary Fig. 2). We found PrL GABAergic projections in the Cg1/2 of SOM^{Cre} but not in PV^{Cre} and VIP^{Cre} mice. Furthermore, we injected CTB into the Cg1/2 of 5HT3A^{EGFP} mice and found retrogradely labeled CTB⁺ / GFP⁺ neurons in the PrL. Based on these results, we conclude that PrL GABAergic projection neurons comprise SOM⁺ and 5HT3A⁺/VIP⁻ neurons.

3. The authors used a viral approach to identify the somatic location of PrL GABAergic cells and their target areas. The presented data related to the presynaptic cells and their axon innervation to other areas in the manuscript are a couple of example images but there is no quantification. Please provide a quantitative analysis of somatic location of projecting GABAergic cells in PrL and their axonal distribution in ACC and other areas. The authors commented these axons are detected in Extended Figure 2 but it is not clear how robust this is. Given PrL and AAC are elongated structures within the frontal cortical area spanning a broad range of AP axis, it will be informative to know the location of presynaptic cells and the areas where these axons innervate.

This question has been raised also by Reviewer 1.

In the original manuscript, we detected cell bodies of PrL GABAergic projection neurons not only in L1, but also in other layers. As the retrograde AAV that we used in the original study was suboptimal (few cells were labeled) and hence precluded a quantitative analysis, we instead injected CTB into the Cg1/2 or the PrL of GAD^{EGFP} mice and performed additional retrograde tracing. Consistent with the results by retrograde AAV, the cell bodies of GABAergic projection neurons were located in all cortical layers as is now shown in Supplementary Fig. 1 and 4.

We also investigated the axonal distribution in more detail, and the quantitative evaluation of the viral tracing experiments is now shown in Figure 1 and Supplementary Figure 3.

4. The authors provided thoughtful comments in the Discussion on the potentially opposite net effect on ACC L5 pyramidal cells by PrL GABAergic cell-mediated inhibition of SBC and NGF in ACC. As the authors pointed out, at least in connection probability and IPSCs, SBC are more strongly connected by PrL GABAergic cells which may lead to inhibition of L5 pyramidal cells. However, current Ca⁺⁺ imaging data show net excitation of L5 pyramidal cells upon vF and optogenetic stimulation. One possibility might be that the spontaneous activity of SBC and NGF are different – NGF may have higher spontaneous activity and be more strongly driven by vF stimulation. Please provide spontaneous activity of SBC and NGF cells in ACC. How the SBC and NGF in ACC are differentially driven by vF-stimulation will be informative to understand how the disinhibition between two different cortical areas by projecting GABAergic cells.

We followed the Reviewer's suggestion, as this indeed might be an explanation, and measured the mean frequency and the mean amplitude of the spontaneous EPSCs in pSBCs and pNGFs in the Cg1/2, but did not find statistical differences (Supplementary Fig. 16). We now mention this in the discussion.

5. Figure 2c: My understanding of the 5HT3R positive GABAergic population in cortex is that SBC and NGF cells are subgroups within the 5HT3R population. Please clarify whether NGF and SBC are 5HT3R negative populations or 5HT3R in the current study indicate non-NGF and non-SBC 5HT3R-positive populations.

The Reviewer must have overlooked that in Figure 2c (now Fig. 2b), the putative 5HT3R+ interneurons we referred to were in L2/3. This was most likely due to the fact that the information was crammed in the middle of the figure. We now place the information more visible in the current version. However, prompted by the Reviewer's comment above, we investigated whether SBCs and NGFs in L1 express 5HT3Rs. To this end we performed experiments entailing local puffing of the 5HT3R agonist, mCPBG. We found that only pSBCs (13 out of 15 cells), but not pNGFs (0 out of 15 cells), were sensitive to mCPBG (Supplementary Fig. 18).

6. Figures 5&6: the authors conducted simultaneous single cell Ca⁺⁺ imaging with optogenetic stimulation. I could not find any description on how the authors controlled the crosstalk between optogenetic stimulation light and scanning light. Please provide clear control data indicating there is no crosstalk.

We feel that a crosstalk between optogenetic stimulation light and scanning light should not constitute a problematic issue for the following reason: Firstly, the fiber plane (L1) and imaging plane (L2/3 and 5) are distinct. Secondly, the wave length for optogenetic stimulation (633 nm) is distinct to that of imaging (920nm). Also, in the previous version of the manuscript, we showed in Extended data Fig. 6 (now Supplementary Fig. 11), that optogenetic stimulation alone does not affect ongoing activity.

Nevertheless, prompted by the Reviewer's concern, we performed cell-attached and whole-cell recordings in ChrimsonR-expressing neurons in PrL and tested the response to two-photon laser scanning (920 nm, same scanning settings and power as those used for Ca²⁺ imaging (Supplementary Fig. 9). Compared to red light stimulation (633 nm), two-photon scanning was unable to activate spiking activity and negligible inward currents (4.0 ± 1.0 pA, $n = 8$ cells) in ChrimsonR-expressing neurons. We hence conclude that the crosstalk is minimal and ignorable.

7. Figure 5: It is not clear why light stimulation reduced the vF stimulation-evoked responses in the control group. Given that AAV-FLEX-tdTomato was expressed in the control group,

the optogenetic stimulation should not evoke any changes. Please provide an explanation on these results.

Given the long experimental time, total 75 minutes, the decrease of vF-evoked responses in the control group may reflect state changes of the head-fixed mice and/or habituation to the vF stimulation. Thus, the decrease in the control group may indicate that the effect in the ChrimsonR group is even larger.

8. Figure 5: The vF stimulation-induced activity of L5 pyramidal cells were still significantly enhanced in session 3 even after an optogenetic stimulation session without light stimulation. Related to this issue, even at a single cell level (Figure 6), vF-activated cells show elevated response even after optogenetic stimulation. Please provide an explanation why the vF stimulation-induced activity is still high in session 3 without optogenetic stimulation. Is it some sort of synaptic plasticity?

Indeed, as inferred by the Reviewer, we also think that the prolonged enhancement of the vF-evoked responses in session 3 may well reflect synaptic plasticity induced in session 2 since the protocol involved repeated application of the stimulus (24 times) in each session. This may lead to a reorganization of the neuronal ensembles involved in cortical processing which is then manifest also in session 3. We did not emphasize this hypothesis because the changes in vF-evoked responses of L5 neurons in sessions 3 relative to session 1 were not significantly increased (Fig. 5k). Nevertheless, there was a tendency indicating prolonged enhancement when considering layer 5 only ($p = 0.054$). When considering the activity in all layers, vF-evoked responses in session 3 were enhanced (Fig. 5i). Interestingly, in the behavioral experiments regarding von Frey testing in mice that received capsaicin injection, we also see prolonged modification in session 3 (Supplementary Fig. 7f). Together these data support the notion that alterations that persist in session 3 may be the result of synaptic plasticity. We have formulated this hypothesis in the discussion.

9. Figure 6: Figure 5 results indicate optogenetic stimulation of PrL GABAergic terminals in ACC only increases L5 pyramidal cell activity during vF stimulation. It is not described whether the single cell analysis in Figure 6 are cells from all layers or from L5. If the data were pulled from all layers, please provide a relationship between ensembles (1 & 2) and layers (L2/3 and 5).

As requested by the Reviewer, we have provided additional analysis of vF-activated and inactivated neurons in L5 (Fig. 6) and L2/3 (Supplementary Fig. 14).

Minor comments

10. In vitro electrophysiology lacks the number of animals that are used in the study. Please report the used animal numbers

We apologize for the omission, and have now reported the number of animals that were used for in vitro electrophysiology in the main text and figure legends.

11. Extended Figure 7b: values for scale bar are missing.

Extended Figure 7b (now Supplementary Fig. 12b) does not show actual responses, but was meant to show schematics of example responses in the withdrawal trials and no-withdrawal

trials. For this reason, there were no numbers for the scale bars. To avoid this misunderstanding, it is now clearly stated in the figure legend that this is a schematic.

12. line 319: Jiang et al., 2015 (PMID: 26612957) can be added here.

Thank you for the suggestion, we have added the reference.

Reviewer #3 (Remarks to the Author):

In this study by Utashiro et al., the authors identified GABAergic projections from the prelimbic (PrL) cortex to the anterior cingulate cortex (ACC), primarily originating from layer (L) 1. The authors investigated which ACC neurons are targeted by PrL and found that the projections are exclusively to L1, particularly to neurogliaform (NGF) cells and single-bouquet cells (SBCs). Furthermore, the authors have evaluated the functional impact of PrL GABAergic projection on ACC circuit revealing that these projections likely regulate L5 pyramidal neurons (PNs) activity via L1 interneurons (INs). The study reveals the function of long-range projection cortico-cortical GABAergic INs, both anatomically and functionally. The results found in the study reveal a different mechanism of top-down control that could be working in parallel (or even competing) with excitatory glutamatergic projections. Additionally, showing a relationship between the investigated projections and different behavioral paradigms is quite interesting and an important contribution to the field of systems neuroscience. However, there are a number of issues that need to be addressed:

1. In Fig. 1

a. It would be nice to quantify the ratio of inhibitory projections to excitatory, i.e. how many of the projections are actually inhibitory in L1. Could be done using a Syn or CamKII promoter based injection in PrL using a different fluorophore and quantifying excitatory vs inhibitory targeting in L1 of ACC.

This is a very interesting question (Reviewer 1 had a similar, but not identical question), and we performed additional experiments that enabled us to provide the density of the GABAergic and glutamatergic projections in each cortical layer. This is now shown in Fig. 1m and n. However, we hesitate to directly compare the densities of GABAergic and glutamatergic projections in L1, mainly for two reasons. 1) AAV-mediated Cre-dependent and CaMKII promoter-dependent gene expression do not guarantee that all GABAergic and glutamatergic neurons at the injection site will be labelled. For instance, the experiment requested by Reviewer 1 (location of source cells) now shown in Supplementary Fig 1g, shows that GABAergic projection neurons of the PrL reside in all cortical layers. We have no way to ascertain that all source cells of the two kinds, i.e. GABAergic and glutamatergic are labeled. 2) Another obstacle preventing a direct comparison resides in the fact that we employed distinct AAVs to label GABAergic and glutamatergic neurons whose efficiency of labelling GABAergic and glutamatergic cells, respectively might be different.

b. Please clarify in the schematics that the animals are GAD-cre

We have added “GAD^{Cre}” to the schematics.

c. Quantify projection density across layers to show L1 specificity in ACC.

We have provided the quantitative analysis. This is now shown in Fig. 1m and n.

d. Fig 1 b, e- it is hard to tell the specificity of the injection to the regions, show an image of the whole section to better appreciate the injection sites in Prl and ACC.

Following the Reviewer's suggestion, we have provided confocal images of the entire section at the injection site.

2. In Fig. 2

a. Raw data examples for the reconstructed morphologies could be nice in supplementary. An important concern is the distinction between NGFs and SBCs. While the authors state that they identify NGFs by first spike latency, they show no quantification for that! FSL should be quantified and be compared between NGFs and SBCs, especially since the data is there! Include criteria for NGF vs SBC classification- intrinsic properties table to be included

More neurons of the two kinds have been filled, and the quantitative evaluation of the axonal density is now also depicted in Fig 1a. As suggested, we summarized the intrinsic properties of pSBCs and pNGFs in a table (Supplementary table 1). For better visualization, we have included a 3D plot showing how the critical parameters were determined (Supplementary Fig. 6).

b. In the discussion the authors claim that NGFs are targeted less than SBCs (line 329), such a claim should be backed by stats in the results, which is not the case. In fact, Fig. 2c shows that its unlikely there's a significant different.

The claim we made was based on the tendency only that NGFs are targeted less. The statistics in fact show that there is no significant difference ($p=0.1361$) as the Reviewer correctly inferred. We hence removed that statement in the discussion, and thank the Reviewer for the thoughtful comment.

c. The authors mention in the methods how they identified SOM and PV INs electrophysiologically but this data is never shown, please include it in supplementary.

As requested, the distinct firing patterns of pSOM⁺, pPV⁺ and also of p5HT3R⁺ INs as well as a 3D plot are now shown in Supplementary Fig. 17.

d. Why are they showing the GABA_AR blocker SR95531 only for the NGFs, was it only tested on NFGs?

Putative SBCs were also tested. We have added an example trace of light-evoked IPSCs recorded in a pSBC in the absence and presence of the GABA_AR blocker, SR, in Fig. 2a.

3. In Fig. 3

a. Include in the schematic the YFP injections in addition

Thank you for noticing this omission, we have added YFP to the schematic.

b. The authors never mentioned the duration of the stimulus, nor in which epoch they stimulate. Not in the figure, not in the main text, and not in the method! Also, what is the

percentage of stimulated trials per session? Please clarify in the text/methods and the schematic.

There is a 50 trial session of sham stimulation (connected to optic fiber but no light delivered) and a 50 trial session of optical stimulation (10-ms pulses at 20 Hz, 473-nm wavelength) where the stimulation is delivered during the entire session. We have clarified this in the methods and have also added it to the results section and the figure legend.

c. In c-e the legend for red is ChR2, however there's no ChR2 in sham! Please clarify what are the sham conditions-is it no light stimulus or a light stimulus of a different frequency?

The red corresponds to the group that received AAV-DIO-ChR2 and the sham condition refers to connection to an optic fiber but no light delivery. Therefore, the sham ChR2 condition is the ChR2 expressing mice connected to an optic fiber but without light delivery. In contrast, the YFP stim condition is YFP expressing mice receiving blue light pulses. We have clarified this in the methods and figure legend and hope that any confusion is now avoided.

d. In the figure caption and methods, the authors mention that there are 3 types of responses, however in the main text they mention 4 types of responses. The authors should specify that trials with responses in the dark are omitted.

Responses in the dark period led to the end of the current trial and 10-20 seconds of darkness at the start of the next trial. We have now stated this more clearly in the methods and results section. Early in the training mice learn not to press during the dark period and therefore in trained animals this is the least frequent response type with many mice making 0% of responses during darkness. This is the reason in some parts of the text we described three 'main' types of responses and did not spend time discussing the very few responses during the dark period. However, for clarity we have now described all four responses in the methods, results and figure legend. In addition, we have reported in the results section that responses in the dark period were not different between YFP and ChR2 in either the low or high demand version of the task.

e. The authors use GAD-Cre mice, hence the GABAergic projections that are stimulated could be coming from anywhere. Understandably, in Fig 1, it was demonstrated that these cells primarily reside in L1, forming a novel L1-L1 connectivity. However, to really show that this circuit is involved in this behavior, the authors should demonstrate this by using a L1 specific Cre line, such as NDNF.

There seems to have been some confusion here. While the targets of PrL GABAergic projections are primarily in L1 (Figure 1), the source cells reside in all layers of the PrL (demonstrated with retrograde tracing, Supplementary Figure 1). Therefore, we are not exclusively looking at L1-L1 connectivity. Whether the projections from different layers have differential effects in the target region is an interesting question but is well beyond the scope of this study.

4. Fig. 4:

a. Would be nice to include a short sentence explaining von Frey filaments in the main text, not everyone might be familiar with it.

We have added a description at the first mention of von Frey filaments in the main text.

b. The schematic for the von Frey filament seems like an afterthought, it is not very informative. Please add more information to understand the experiment

In order to try and make the schematic more informative, we have altered it to hopefully show more clearly that the stimulation lasts throughout the second session. We are unclear as to what further information the schematic should convey. The three test sessions, the experimental time course and session containing laser stimulation are all indicated. If the reviewer has suggestions, we would be happy to implement them.

c. Show a comparison of the von Frey response across sessions in both control and optogenetic conditions

*We are not sure whether the reviewer would like to see this in addition to the responses split by session, or as a replacement to the already presented plots. Showing the responses across sessions addresses the issue as to whether the responses are stable across the three testing sessions or whether there is some tolerance or even sensitization to the repeated von Frey filament applications. However, while potentially interesting, this is not the main focus of the experiments - the statistical analysis is performed within each session which corresponds to the layout in the presented plots. We have generated a plot with all three sessions for YFP and ChR2 and have shown it (Fig. 4 for Reviewer 3). However, given that the plots are normally split by testing session (e.g. Tan, L., Pelzer, P., Heintz, C. et al. A pathway from midcingulate cortex to posterior insula gates nociceptive hypersensitivity. *Nat Neurosci* 20, 1591–1601 (2017)) we have left them as such in the manuscript. We would of course include this as a supplementary figure if the reviewer insists, however as we already have a considerable number of supplementary figures and as this one does not address a direct question raised by the experiments, for now we have decided not to include it.*

5. Fig. 5: in this figure and figure 6 and related supps axis have very small font! You can barely read it!

According to the publication guideline (font size, 5-7 p), we used font size 5 or 6 in the original figures. Following the Reviewer's suggestion, we have used font size 7 in the new figures.

a. Demonstrate that the grin lens implantation does not cause disruption of L1 fibers. Do a quantification and include images of the brains after perfusion to show the intactness of L1 in these experiments.

We here provide a schematic figure (Fig. 3 for Reviewer 2 and Reviewer 3) to show that the intactness of L1 is not a concern for the following reason. The location of the ACC (Cg1/2) is such that the cortical layers in the two hemispheres are right and left to the midline and L1 is directly ventral to the midline sinus. As shown in the schematic, the GRIN lens is implanted lateral to the sinus such to avoid injuring it, which also places the lens lateral to L1 of the ACC (Cg1/2). This guarantees also the intactness of L1. This information has also been added to the Methods section. Thank you for drawing our attention to this issue. It is important to convey this information to the reader.

b. Why was 0.6gms used in these experiments? According to fig 4, 0.6gms had no difference with and without optogenetic stimulation. Please clarify.

The head-fixed mice used in Ca²⁺ imaging experiments demonstrated less withdrawal behavior (Supplementary Fig. 12c) compared to freely-moving mice (Fig. 4) such that there was no response difference with 0.4-g yet, but with 0.6-g filaments a difference was seen. Based on this, we used 0.6-g filaments in the Ca²⁺ imaging experiments.

We think that the difference in threshold between the von Frey experiments with and without Ca²⁺ imaging may well result from the differential experimental setting (in particular head fixation and implantation). Thus, for the Ca²⁺ imaging experiments, mice were head-fixed under the microscope restricting the movements of the mice compared to that of freely moving animals. In addition, the chronic implantation with a GRIN lens and an optic fiber is very likely to affect behavior. A previous study (Tan et al., Nat. Commun., 2019) showed that mice chronically implanted with cortical microelectrodes exhibited less withdrawal behavior than mice without implantation. This information has been added to the Methods section.

c. The schematic is not very informative with respect to the results- when was the optogenetic stimulation delivered? Elaborate on the timing within the schematic. This goes for all the figures, it is not very clear when/how long the stimuli lasted.

Following the Reviewer's suggestions, to exemplify the details of the stimulation better, we updated the schematic of the procedure used for the Ca²⁺ imaging experiments, and hope that this is now clearer (Fig. 5e).

d. Please put comparisons and level of decrease for each in the main text (numerical values and statistics).

As requested by the Reviewer, the numerical values and statistics have been added also in the main text.

e. For L2/3 and L5 responses, show example traces. From the confocal image example, there doesn't seem to be any expression of GCaMP6 in L2/3. How many cells were imaged in L2/3 vs L5. Could the no difference be merely due to low number of imaged cells?

We have provided example traces of the responses from L2/3 (Supplementary Fig. 14c,g) and L5 (Fig. 6d,h).

As shown in Fig. 1 k and o, GABAergic projections are denser in the superficial part of L1 (i.e. L1a) and there is a "gap" between L1a and L2/3 (this is in fact layer L1b). This led the Reviewer to believe that there is no expression of GCaMP6 in L2/3 in the original confocal image (Fig. 5c), but in fact, the confocal image that we showed in the original manuscript comprised L1 and L2/3, and CaMP6-expressing cells were in L2/3. We provide this panel again so the Reviewer can verify our assertion (Fig. 5 for Reviewer 3), and now the border between L1 and L2/3 is indicated. However, for lack of space (as several requested panels had to be added to the revised version) and to avoid confusion, we removed this panel in the revised version. In addition, we have provided the exact cell numbers in Fig. 5, Fig. 6 and Supplementary Fig. 14.

f. What about responses differences in L1 cells themselves? Does vF cause an inhibition in these cells? In order to decipher circuit mechanism of this phenomenon the authors should also record responses of L1 INs themselves.

This is an interesting question, but to record responses of L1 INs in vivo is beyond the scope of this study. It would entail experiments lasting at least as long as this entire study (several years). Our main question was not so much what the different behaviors, including von Frey stimulation, do to L1 INs, but what modulation of L1 INs via long-range GABAergic input does both at the network/neuronal and behavioral level.

We thought, however, that it might be informative to show at least in one of our behavioral experiments that L1 INs are recruited and that the recruitment is stimulus-dependent. To this end, we investigated cFos expression in control mice and in mice that received von Frey stimulation subsequent to capsaicin injection. We found a significant increase in the number of cFos expressing INs in the latter cohort. This data is shown in Supplementary Fig. 7a,b.

Minor points

It appears the manuscript was written by multiple people- which makes the writing style different in different parts of the manuscript. Please unify the writing styles as well as ensure the manuscript has been proofread for grammatical errors/usage

I (the PI) am aware of the potential problem that may arise when several authors are involved writing a manuscript. Not only I, but also the three first-coauthors read the manuscript carefully, seeking to arrive at a homogenous and balanced manuscript. One of the authors (Duncan MacLaren) is a native English speaker, and all of us went again through the revised version of the manuscript. Should the Reviewer think of anything specific that he/she recommends to change, we are happy to change this.

a. The authors should address every subpanel for every figure in the main text. Figure 3 has 5 panels but only referenced in the main text as Fig. 3

This has been corrected.

b. In Fig 6, use a different color (not red) to highlight vF activated neurons. At the moment it is confusing to use red since previously it denoted the chrimson expressing ones

We followed the Reviewer's advice and changed the color from red to orange to indicate vF-activated neurons in Fig. 6, Supplementary Fig. 13 and Supplementary Fig. 14.

c. In extended data Fig. 6. The authors should compare control vs chrimson rather than chrimson in different groups! Also they should compare optogenetic activity and baseline activity in small windows of time at a single neuron level rather than a population and averaging over a whole sessions! Using these analysis, if there was a crosstalk it will obviously average out!

In the extended data Fig. 6 the comparison must be performed this way because a comparison between control vs. chrimson would entail cells in different mice. This control experiments should serve only the purpose to demonstrate that optogenetic stimulation per se does not affect neuronal activity in mice subjected to the specific conditions that are shown. However, we performed control experiments as envisaged by the Reviewer (Supplementary

Fig. 9), but these had to be performed in vitro. We show in this figure that ChrimsonR-expressing neurons are activated at 633 nm, but not at 920 nm, i.e. the wavelength that is used for calcium imaging.

d. Calling blocks before, during, and post optogenetic “sessions” is very confusing, as in between sessions implies to me that the animals were returned to animal facility. But I guess this is personal point of view.

We preferred the term “session” because, at least in behavioral experiments (Y maze, Morris water maze etc.), we and others use “blocks” when the conditions are identical. Conversely, we found the term “sessions” when the conditions changed (for instance in papers reporting on von Frey tests before, during or after capsaicin injection; Tan et al., Nat. Commun., 2019). Since we used the terminology for the behavioral tests, we felt that it is best to stick to this terminology also when referring to the experiments involving optogenetic manipulations. We hope that the improved schematics of the different experimental paradigms help to prevent confusion.

e. A reference should be cited in line 314 (... their recruitment shapes the activity of local circuits).

Following the Reviewer’s suggestions, we have added the following references: Anastasiades et al., Neuron, 2021, Jiang et al., Nat. Neurosci., 2013, Lee et al., Cereb. Cortex., 2015, Schuman et al., Annu. Rev. Neurosci. 2021

f. In line 318 authors should state what they mean by distinctly wired, and why this wiring is important to their findings.

The two papers that we cited describe the complex wiring of NGFs and SBCs with downstream neurons, and predict an opposite effect of L5 output neurons. We rephrased the sentence such to emphasize this point without overburdening the reader with details.

g. In line 320 , the use of the word “master-regulators” is too strong. They are gatekeepers, and control the flow of excitation, but master regulators implies something that is very broad and controls every aspect of information processing.

Thank you for the suggestion that we have implemented, we agree that it does indeed convey the message better and correctly.

h. In line 321 the authors claim that these projections control neuronal signal across layers, but also in the result they claim (and show) that the control is specific to L5 and not L2/3. Please clarify?

We may not have phrased our statement clearly enough, but what we meant was that, by projecting at long distance to L1 INs, PrL GABAergic projection neurons control/modulate the activity of neuronal networks in the target region. “Across cortical layers” meant the circuits between L1 and L5 in the Cg1/2. We have rephrased the sentence, and hope that, in conjunction with the other sentences of that paragraph, the message is conveyed more precisely in the revised version.

i. The authors mention in the results injection in medial orbital cortex (MO; supp. 2) but in the methods no injection site is mentioned for MO. If it was off target injections of PrL there's no harm in mentioning that, perhaps mentioning the depths of these off-target injections could save animals in other labs!

We cannot answer the Reviewer's question for the following reason: Previous injections were performed by a doctoral student (i.e. author Birgit Wojak) who is no longer in the lab. Prompted by the Reviewer's comment, we (first author Nao Utashiro) performed additional injections (n = 3 mice) using the same coordinates as before. All new injections turned out to be confined to the PrL (coordinates are mentioned. Differences from previous results might be experimenter related (the retraction of the injection needle is a very crucial variable accounting for differences in the injected volume), batch of the virus, etc). We decided not to expand the anatomical investigations to include MO as this would have needed additional injections confined to the MO (coordinates would have to be established first). As restricted injection into the MO is not trivial and would have necessitated the injection of quite some animals (for the revision of this manuscript we have used already a total of 69 mice), we removed this panel.

j. Overall, with so many injections with different viral vectors x transgenic animals and different injection volumes, and different brain areas, would be nice to have supplementary summary table

Following the Reviewer's advice, we have provided the requested information which is now shown in Supplementary table 3.

Reviewer #4 (Remarks to the Author):

The authors identified GABAergic projections connecting two parts of the mouse medial prefrontal cortex (mPFC), namely a part they call prelimbic cortex (PrL) and a part they call anterior cingulate cortex (ACC). They also identified the neurons targeted by these projections as being single-bouquet cells and neurogliaform cells. Furthermore, they performed a functional characterization of

The authors use anterograde and retrograde tracing and identified reciprocal GABAergic projections between PrL and ACC. They found layer I to be the mainly targeted layer. They performed patch clamp recordings to identify the cells in ACC being targeted by the PrL GABAergic efferents, and vice versa. In both areas, the GABAergic projections selectively inhibited putative single-bouquet and neurogliaform cells. Furthermore, the authors determined that stimulation of PrL GABAergic projections in the ACC resulted in an increased impulsivity and affected the performance of mice in an operant test of sustained attention. Additionally, the authors determined that optogenetic activation of these projections increased sensitivity and response of mice to mechanical stimulation. The authors also determined that recruitment of PrL GABAergic terminals in the ACC enhanced the activity of layer V neurons.

These excellent results are unfortunately accompanied by a deplorably low degree of anatomical accuracy which must be corrected not only for this study to really advance our understanding of medial prefrontal circuitry, but also to enable reproducibility of the results.

The authors state that the human and mouse mPFC comprise several subregions, including PrL and ACC. The term prelimbic cortex is used in rodent, but not primate research. However, the authors use it indistinctively for both species. Specifically, PrL designates an

architecturally distinct area (not a subregion or region) in the rodent brain considered to be homologous to area 32 in the brains of humans and non-human primates (Preuss et al., 1995). The term ACC is used in functional imaging studies to designate a region of the human cingulate cortex encompassing Brodmann's areas 24, 25, 32 and 33. Given the drawings provided in Fig. 1a and 1d, the area designated as ACC by the authors actually corresponds to the portion of cortex occupied by the dorsal anterior cingulate area (ACd, or area 24d) and the ventral anterior cingulate area (ACv, or area 24v) of Preuss et al. (1995). And the injection site could actually be at the border between these two areas. Areas ACd and ACv correspond to areas cg1 and cg2 of Franklin and Paxinos (2013), respectively. Thus, the term ACC is not synonym of ACd and ACv. Rather, ACC encompasses areas IL, PrL, cg1 and cg2. Furthermore, areas cg1 and cg2 have each been subdivided into a rostral (cg1'/cg2') and a caudal (cg1''/cg2'') area (Vogt & Paxinos, 2014; van Heukelum et al., 2020).

We are grateful that this Reviewer appreciates our study and regret that he/she is not happy with the anatomical nomenclature that we used. We followed the terminology that was used for the rat brain (Hoover & Vertes, Brain Struct. Funct. 2007, Anastasiades & Carter, Trends Neurosci. 2021), and are somewhat at a loss what to do at this point. We had adopted the nomenclature frequently used for rodents, in particular in the context of behavioral or functional studies. We would like to point out just a few (Barthas et al., Biol Psychiatry., 2015, Ong et al., Mol. Neurobiol., 2019, Kummer et al., Int. J. Mol. Sci., 2020, Seamans et al., Behav. Neurosci., 1995, van der Veen et al. Commun Biol. 2021, Anastasiades & Carter, Trends Neurosci. 2021).

To allow identification of areas regardless of nomenclature, we followed the Reviewer's advice and have provided confocal images of the entire section at the injection site (Fig. 1d-f, Supplementary Fig. 1b, Supplementary Fig. 3d-f, and Supplementary Fig. 4b). However, prompted by the Reviewer's comments, and in particular in the attempt to minimize further confusion in the field, we have now provided, additional information with the pertinent references in the introduction, based on which the reader's attention is drawn to the fact that species-specific nomenclatures have been often used. There we also mention the homologous areas with frequently used nomenclatures in rodents vs. primates. In the manuscript we now consistently refer to PrL and, instead of ACC, to Cg1/2. We hope the Reviewer is convinced by the correctness of these assertions based on the figures that we now provide (Fig. 1d-f, Supplementary Fig. 1b, Supplementary Fig. 3d-f, and Supplementary Fig. 4b).

In which of these four areas (cg1, cg2, cg1', cg2') did the authors inject the retrograde tracer? This must be specified in order to enable a correct interpretation of the results. Ideally, the authors should provide a series of overview images of sections containing the injection site. These could be either a low-resolution fluorescent image of the entire section, or a bright-field or dark-field image of the sections in question (as was done, e.g., in Fig. 7 of Király et al., 2020). These images should be accompanied by high resolution micrographs of sections enabling architectonic identification of the site.

The retrograde tracer CTB was injected at the border between Cg1 and Cg2. As requested, we have provided confocal images of the whole section (Supplementary Fig. 1b). We also made a figure for this Reviewer (Fig. 6, 7 for Reviewer 4) comprising a series of overview images of sections containing the injection site in the Cg1/2 and the PrL to further substantiate our claim.

Is the area they designate PrL really area PrL? Could it be part of area M2? In figure 1d area PrL clearly extends onto the dorsal surface of the hemisphere, although this area is generally restricted to the mesial surface of the hemisphere. Also here, the authors should provide overview images of sections containing the injection site and high resolution micrographs of sections enabling architectonic identification of the site.

We are convinced that both the anterograde and retrograde tracing supports the notion that the designation of PrL is correct. As stated above, we followed the Reviewer's suggestion and have provided the confocal images of the entire section at the injection site (Fig. 1d-f). The center of the injection sites was within the PrL, although some minimal spillover of the anterograde tracer to some individual cells cannot be excluded with certainty. But the retrograde tracing corroborates the results of the anterograde tracing.

Minor points:

I find it surprising and an example of poor citation practice that the authors have not cited a single article by Brent Vogt, one of the neuroanatomists who has most contributed to our understanding of the architectonic segregation of the cingulate cortex across species.

We acknowledge that our focus was the rodent brain, and admit that we were really not much aware of the literature regarding the cross-species differences. We apologize for this and thank the Reviewer for having drawn our attention to this issue of debate. We hope to have done a better job in the revised version of the manuscript containing the pertinent literature.

A table with information concerning the exact number of animals included in each experimental group would be helpful.

We have provided a supplemental table containing the requested information.

Refs.

Franklin & Paxinos (2013) The mouse Brain in Stereotaxic Coordinates. Amsterdam, Academic Press

Király et al. (2020) Nat Commun 11: 4686. <https://doi.org/10.1038/s41467-020-18472-y>

van Heukelum et al., (2020) TINS 43: 285-299

Vogt & Paxinos (2014) Brain Struct Funct 219: 185-192

Fig. 1 for Reviewer 1 Morphological reconstructions of 13 L1 INs in the Cg1/2

a, Reconstructions from 6 pSBCs. Somata and dendrites are indicated in black. Axonal arborization is indicated in green and blue. The borders of cortical layers are indicated as dotted lines. b, Reconstructions from 7 pNGFs.

Fig. 2 for Reviewer 1 Interaction plots of vF-responses of L2/3 and L5 neurons

a, Area under the curves (AUCs) of the vF-evoked responses of L2/3 vF-activated neurons in control and in ChrimsonR mice shown as box plots and **b**, as interaction plots. **c**, Area under the curves (AUCs) of the vF-evoked responses of L2/3 vF-inactivated neurons in control and in ChrimsonR mice shown as box plots and **d**, as interaction plots. **e**, Area under the curves (AUCs) of the vF-evoked responses of L5 vF-activated neurons in control and in ChrimsonR mice shown as box plots and **f**, as interaction plots. **g**, Area under the curves (AUCs) of the vF-evoked responses of L5 vF-inactivated neurons in control and in ChrimsonR mice shown as box plots and **h**, as interaction plots. Circles in interaction plots indicate values from

individual cells. n.s., not significant, *P < 0.05, **P < 0.01, ***P < 0.001, Friedman test and Wilcoxon signed-rank test followed by Bonferroni correction.

Fig. 3 for Reviewer 2 and Reviewer 3 L1s of the Cg1/2 are located below the superior sagittal sinus.

- a**, Schematic of mouse brain showing superior sagittal sinus. Red lines indicate blood vessels.
b, Coronal schematic of the indicated area in panel (a). The Cg1/2 are located to the right and left of the midline of the two hemispheres. **c**, Magnified schematic of the boxed area in panel (b). GRIN lens is implanted to avoid the sinus above L1 of the Cg1/2.

Fig. 4 for Reviewer 3 Comparison of the von Frey responses across sessions in the control and ChR2 groups.

The data from fig. 4 of the main text plotted on a single graph. This enables the direct comparison of the pre, peri and post stim conditions and comparison between the YFP control group and ChR2 optogenetic group. The post stim responses are not different to the pre stim responses (all p values > 0.10 , repeated measures ANOVA with Tukey corrected multiple comparisons) indicating that there is no habituation or sensitization to the repeated von Frey applications. In the pre and post stim conditions there are also no differences between the YFP and ChR2 groups, confirming that there are no pre-existing differences between the experimental groups (all p values > 0.10). For the 0.16g and 0.4g conditions (which in the main figure are significantly different between YFP and ChR2) ChR2 pre vs. peri and ChR2 peri vs. post are $p < 0.05$.

Fig. 5c

Fig. 5 for Reviewer 3 GCaMP6s was expressed in L2/3 neurons.

a, Confocal image which was shown in Fig. 5c of the original figure. Representative confocal image of Cre-dependent ChrimsonR-tdTomato expression in the axons of PrL GABAergic projections (magenta) and CaMKII-promoter-dependent GCaMP6s (green) expression in the Cg1/2. Scale bar, 100 μ m. **b**, Confocal image showing the axons of PrL GABAergic projections (magenta) and DAPI signals to indicate the border of L1 and L2/3.

Fig. 6 for Reviewer 4 CTB injection sites was at the border of the Cg1 and the Cg2.

Confocal images of a DAPI stained coronal section showing the CTB-Alexa 555 signal (magenta) at the injection site in the Cg1/2 from anterior (i) to the posterior (iv). a, Mouse

brain atlas (Franklin & Paxinos, Third Edition (2007)) was fitted to coronal sections based on anatomical landmarks identified in the confocal images. **b**, Confocal image shown in **(a)** without an atlas. **c**, Brighter image shown in **(b)**. Scale bar, 500 μm , Abbreviation, PrL, prelimbic cortex, Cg1/2, cingulate area 1 and 2, IL, infralimbic cortex, M1, primary motor cortex, M2, secondary motor cortex

Fig. 7 for Reviewer 4 CTB was injected within the PrL.

Confocal images of a DAPI stained coronal section showing the CTB-Alexa 555 signal (magenta) at the injection site in the PrL from anterior (i) to the posterior (iv). a, Mouse brain atlas (Franklin & Paxinos, Third Edition (2007)) was fitted to coronal sections based on

anatomical landmarks identified in the confocal images. **b**, Confocal image shown in **(a)** without an atlas. **c**, Brighter image shown in **(b)**. Scale bar, 500 μm , Abbreviation, PrL, prelimbic cortex, MO, medial orbital cortex, VO, ventral orbital cortex, LO, lateral orbital cortex, DLO, dorsolateral orbital cortex, FrA, frontal association cortex, Cg1, cingulate area 1, M2, secondary motor cortex

REVIEWERS' COMMENTS

Reviewer #1 (Remarks to the Author):

The authors have completely addressed my concerns, and I would like to congratulate them on an important and comprehensive study dissecting the organization and function of long-range inhibition in the mPFC.

I append below a few suggestions they may wish to take into account when preparing the final version of the paper.

- Figure 1 for Reviewer 1 is quite useful, so I suggest appending it to the manuscript

- Figure 2a: The axonal density plots are hard to read, I suggest enlarging the y-axis

- Figure 5g: Session 2 displays lower AUC values than session 1, and they interpret this as potentially due to synaptic plasticity. However, if that were the case one would predict the effect to be observed also in session 3. I submit that this may just be an acute effect of the light delivery

- Fig. 6b and c, ED Fig. 13 & ED Fig. 14: it might make sense to indicate in the heat map when the optogenetic stimulation is delivered. Also, it could be helpful to sort the data based on either the magnitude of the responses or the time that it takes to reach a maximal response

Reviewer #2 (Remarks to the Author):

In this version, the authors worked to address my concerns in detail, adding clarification for some of the methodological issues, as well as refining the data presentation.

1. The activity of GABAergic neurons in PrL and Cg1/2 during vF stimulation and/or a visual attention task: Due to technical difficulty, the authors indirectly measured the GABAergic neuronal activity by

quantifying cFos expression. This data indirectly provides evidence that L1 GABAergic cells are active during the task. Since the authors already have this data, please provide the analysis of cFos expression in projecting GABAergic cells in PrL.

2. Cell type of projecting GABAergic cells in PrL: The authors provided convincing tracing data to demonstrate that the projecting cells are SOM+ and 5HT3A+/VIP- cells. It would be informative to present PVcre and VIPcre data together in Supplementary Fig. 2.

3. The laminar location of PrL GABAergic cells: The authors provided quantitative analysis on the somatic laminar position and axonal distribution.

4. Spontaneous and driven activity of SBC and NGF cells in Cg1/2: The mean frequency and amplitude of spontaneous EPSCs from SBCs and NGFs in slice recording may not directly address the spontaneous and driven activity of SBC and NGF cells in vivo. However, I understand that studying the activity of SBC and NGF cells in vivo could be another project.

5. SBC cells, NGF cells, and 5HT3R positive GABAergic cells: The authors clarified this issue.

6. Crosstalk between optogenetic stimulation and scanning: The authors provided clear evidence addressing the crosstalk.

7. Decreased activity of PrL GABAergic projecting neurons in the control group: The authors' response to the decreased activity in the control group is not clear to me. However, I do agree that the optogenetic activation of PrL GABAergic inputs to Cg1/2 enhances the evoked responses in L5 cells in Cg1/2.

8. Synaptic plasticity from GABAergic projecting cells in PrL to L5 pyramidal cells in Cg1/2: It is an interesting question to address the mechanism of synaptic plasticity, but I understand this could be another study.

9. The activity of L5 pyramidal cells in Cg1/2: The new analysis is informative to understand the different activity patterns of L5 pyramidal cells in Cg1/2. Please provide what percentage of L5 recorded neurons show significant changes upon optogenetic and vF stimulation.

Minor comments were appropriately addressed.

Reviewer #3 (Remarks to the Author):

The authors have addressed all my concerns. No further comments.

Reviewer #4 (Remarks to the Author):

The authors have satisfactorily addressed all my concerns

REVIEWERS' COMMENTS

Reviewer #1 (Remarks to the Author):

The authors have completely addressed my concerns, and I would like to congratulate them on an important and comprehensive study dissecting the organization and function of long-range inhibition in the mPFC.

I append below a few suggestions they may wish to take into account when preparing the final version of the paper.

- Figure 1 for Reviewer 1 is quite useful, so I suggest appending it to the manuscript

We followed the Reviewer's suggestion and have included a new Supplementary Figure (Supplementary Figure 8).

- Figure 2a: The axonal density plots are hard to read, I suggest enlarging the y-axis

We have enlarged the y-axis as suggested.

- Figure 5g: Session 2 displays lower AUC values than session 1, and they interpret this as potentially due to synaptic plasticity. However, if that were the case one would predict the effect to be observed also in session 3. I submit that this may just be an acute effect of the light delivery

We carefully checked the paper and the answers to all Reviewers, and believe that this is a misunderstanding. We mentioned plasticity only in the context of the augmented response that persists in session 3 following stimulation in session 2 in ChrimsonR mice (Figure 5h). The diminished response in session 2 in Control mice (Figure 5g) we cannot explain, but can only speculate. As we indicated in our answer to question 7 of Reviewer 2, this might reflect a state change and/or habituation.

-Fig. 6b and c, ED Fig. 13 & ED Fig. 14: it might make sense to indicate in the heat map when the optogenetic stimulation is delivered. Also, it could be helpful to sort the data based on either the magnitude of the responses or the time that it takes to reach a maximal response

As requested, we have added the ticks indicating when the optogenetic stimulation is delivered. The data were sorted based on the maximum value during vF stimulation in session 1. This information was not in the figure legend, but has been added now.

Reviewer #2 (Remarks to the Author):

In this version, the authors worked to address my concerns in detail, adding clarification for some of the methodological issues, as well as refining the data presentation.

1. The activity of GABAergic neurons in PrL and Cg1/2 during vF stimulation and/or a visual attention task: Due to technical difficulty, the authors indirectly measured the GABAergic neuronal activity by quantifying cFos expression. This data indirectly provides evidence that L1 GABAergic cells are active during the task. Since the authors already have this data, please provide the analysis of cFos expression in projecting GABAergic cells in PrL.

Unfortunately, these experiments solely answer the question that L1 INs in the target area (Cg1/2) are activated during vF stimulation, and also that there is differential activation in control and capsaicin injected mice (Supplementary Figure 9a and b). In other words, we quantified the putative target cells, but the experiments do not contain the information that the Reviewer is interested in. Since we have no specific molecular marker for GABAergic projection neurons, we cannot investigate these brains post hoc, as cFos+/GAD+ cell in the PrL would also include all interneurons. In other words, to answer the question whether the GABAergic projection neurons themselves express cFos would necessitate new experiments with prior retrograde labeling. But even so, it might be difficult to discern the GABAergic projecting neurons (that are a minority dispersed in all layers) from the much larger population of GABAergic interneurons.

2. Cell type of projecting GABAergic cells in PrL: The authors provided convincing tracing data to demonstrate that the projecting cells are SOM+ and 5HT3A+/VIP- cells. It would be informative to present PVcre and VIPcre data together in Supplementary Fig. 2.

As suggested, these data are now included in Supplementary Figure 2e-h and 3a-d.

3. The laminar location of PrL GABAergic cells: The authors provided quantitative analysis on the somatic laminar position and axonal distribution.
4. Spontaneous and driven activity of SBC and NGF cells in Cg1/2: The mean frequency and amplitude of spontaneous EPSCs from SBCs and NGFs in slice recording may not directly address the spontaneous and driven activity of SBC and NGF cells in vivo. However, I understand that studying the activity of SBC and NGF cells in vivo could be another project.
5. SBC cells, NGF cells, and 5HT3R positive GABAergic cells: The authors clarified this issue.
6. Crosstalk between optogenetic stimulation and scanning: The authors provided clear evidence addressing the crosstalk.
7. Decreased activity of PrL GABAergic projecting neurons in the control group: The authors' response to the decreased activity in the control group is not clear to me. However, I do agree that the optogenetic activation of PrL GABAergic inputs to Cg1/2 enhances the evoked responses in L5 cells in Cg1/2.
8. Synaptic plasticity from GABAergic projecting cells in PrL to L5 pyramidal cells in Cg1/2: It is an interesting question to address the mechanism of synaptic plasticity, but I understand this could be another study.
9. The activity of L5 pyramidal cells in Cg1/2: The new analysis is informative to understand the different activity patterns of L5 pyramidal cells in Cg1/2. Please provide what percentage of L5 recorded neurons show significant changes upon optogenetic and vF stimulation.

This question might be interesting if the analysis had been performed in a different way. But as we included only neurons that exhibited an altered response in any of the three sessions (the non-responsive neurons are already excluded), we feel that this information is not pertinent to the data analysis that homes in on the analysis of the same neurons in the three sessions.

Minor comments were appropriately addressed.

Reviewer #3 (Remarks to the Author):

The authors have addressed all my concerns. No further comments.

Reviewer #4 (Remarks to the Author):

The authors have satisfactorily addressed all my concerns